# In-distribution and Out-of-distribution Generalization for Graph Neural Networks

## Abstract

Graph neural networks (GNNs) are models that allow learning with structured data of varying size. Despite their popularity, theoretical understanding of the generalization of GNNs is an under-explored topic. In this work, we expand the theoretical understanding of both in-distribution and out-of-distribution generalization of GNNs. Firstly, we improve upon the state-of-the-art PAC-Bayes (in-distribution) generalization bound primarily by reducing an exponential dependency on the node degree to a linear dependency. Secondly, utilizing tools from spectral graph theory, we prove some rigorous guarantees about the out-of-distribution (OOD) size generalization of GNNs, where graphs in the training set have different numbers of nodes and edges from those in the test set. To empirically verify our theoretical findings, we conduct experiments on both synthetic and real-world graph datasets. Our computed generalization gaps for the in-distribution case significantly improve the state-of-the-art PAC-Bayes results. For the OOD case, experiments on community classification tasks in large social networks show that GNNs achieve strong size generalization performance in cases guaranteed by our theory.

## 1 Introduction

Graph neural networks (GNNs), firstly proposed in Scarselli et al. (2008), generalize artificial neural networks from processing fixed-size data to processing arbitrary graph-structured or relational data, which can vary in terms of the number of nodes, the number of edges, and so on. GNNs and their modern variants (Bronstein et al., 2017; Battaglia et al., 2018) have achieved state-of-the-art results in a wide range of application domains, including social networks (Hamilton et al., 2017), material sciences (Xie & Grossman, 2018), drug discovery (Wieder et al., 2020), autonomous driving (Liang et al., 2020), quantum chemistry (Gilmer et al., 2020), and particle physics (Shlomi et al., 2020).

Despite their empirical successes, the theoretical understanding of GNNs are somewhat limited. Existing works largely focus on analyzing the expressiveness of GNNs. In particular, Xu et al. (2018) show that GNNs are as powerful as the Weisfeiler-Lehman (WL) graph isomorphism test (Weisfeiler & Leman, 1968) in distinguishing graphs. Chen et al. (2019) further demonstrate an equivalence between graph isomorphism testing and universal approximation of permutation-invariant functions. Loukas (2019) show that GNNs with certain conditions (*e.g.*, on depth and width) are Turing universal. Chen et al. (2020) and Xu et al. (2020a) respectively examine whether GNNs can count substructures and perform algorithmic reasoning.

In the vein of statistical learning theory, generalization analyses for GNNs have been developed to bound the gap between training and testing errors using VC-dimension (Vapnik & Chervonenkis, 1971), Rademacher complexity (Bartlett & Mendelson, 2002), algorithmic stability (Bousquet & Elisseeff, 2002), and PAC-Bayes (McAllester, 2003) (a Bayesian extension of PAC learning (Valiant, 1984)). Depending on whether the problem setup is *in-distribution* (ID) or *out-of-distribution* (OOD), *i.e.*, whether test data comes from the same distribution as training data, we categorize the literature into two groups.

**ID Generalization Bounds.** Scarselli et al. (2018) provide a VC-dimension based generalization bound for GNNs whereas Verma & Zhang (2019) present the stability-based generalization analysis for single-layer graph convolutional networks (GCNs) (Kipf & Welling, 2016). Both consider node classification and assume the node features are independent and identically-distributed (IID), which conflicts with the common relational learning setup (*e.g.*, semi-supervised node classification) at which GNNs excel. Relying on the neural tangent kernel (NTK) approach (Jacot et al., 2018), Du et al. (2019) characterize the generalization bound of infinite-width GNNs on graph classification. Garg et al. (2020) derive the Rademacher complexity based bound for message passsing GNNs on graph classification. Lv (2021) establish results for GCNs on node classification using Rademacher complexity as well. Based on PAC-Bayes, Liao et al. (2020) obtain a tighter bound for both GCNs and message passsing GNNs on graph classification compared to (Garg et al., 2020; Scarselli et al., 2018). Subsequently, Ma et al. (2021) also leverage PAC-Bayes and show generalization guarantees of GNNs on subgroups of nodes for node classification. More recently, Li et al. (2022) study the effect of graph subsampling in the generalization of GCNs.

**OOD Generalization** Yehudai et al. (2021) study *size generalization* for GNNs — this is a specific OOD setting where training and testing graphs differ in the number of nodes and edges. They show negative results that specific GNNs can perfectly fit training graphs but fails on OOD testing ones. Baranwal et al. (2021) consider specific graph generative models, *i.e.*, the contextual stochastic block model (CSBM) (Deshpande et al., 2018), where CSBMs during training and testing are of the same means but different number of nodes, intra-, and inter-class edge probabilities. They present generalization guarantees for single-layer GCNs on binary node classification tasks. Later, Maskey et al. (2022) assume yet another class of graph generative models, *i.e.*, graphons, where the kernel is shared across training and testing but the number of nodes and edges could vary. They obtain generalization bounds of message passing GNNs on graph classification and regression that depend on the Minkowski dimension of the node feature space. Relying on a connection of over-parameterized networks and neural tangent kernel, Xu et al. (2020b) find that task-specific architecture/feature designs help GNNs extrapolate to OOD algorithmic tasks.

Wu et al. (2022a) propose explore-to-extrapolate risk minimization framework, for which the solution is proven to provide an optimal OOD model under the invariance and heterogeneity assumptions. Yang et al. (2022) propose a two-stage model that both infers the latent environment and makes predictions to generalize to OOD data. Empirical studies suggest it works well on real-world molecule datasets. Wu et al. (2022b) study a new objective that can learn invariant and causal graph features that generalize well to OOD data empirically. All above works follow the spirit of invariant risk minimization (Arjovsky et al., 2019) and focus on designing new learning objectives. Instead, we provide generalization bound analysis from the traditional statistical learning theory perspective.

**Our Contributions.** In this paper, we study both in-distribution and out-of-distribution generalization for GNNs. For in-distribution graph classification tasks, we significantly improve the previous state-of-the-art PAC-Bayes results in (Liao et al., 2020) by decreasing an exponential dependency on the maximum node degree to a linear dependency. For OOD node classification tasks, we do not assume any known graph generative models which is in sharp contrast to the existing work. We instead assume GNNs are trained and tested on subgraphs that are sampled via random walks from a single large underlying graph, as an efficient means to generate a connected subgraph. We identify interesting cases where a graph classification task is theoretically guaranteed to perform well at size generalization, and derive generalization bounds. We validate our theoretical results by conducting experiments on synthetic graphs, and also explore size generalization on a collection of real-world social network datasets. In the in-distribution case, we observe an improvement of several orders of magnitude in numerical calculations of the generalization bound. In the out-of-distribution case, we validate that, in cases where the theory guarantees that size generalization works well, the prediction accuracy on large subgraphs is always comparable to the accuracy on small subgraphs, and in many cases is actually better.

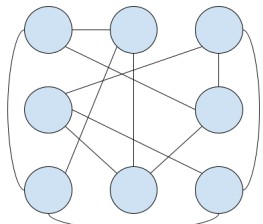 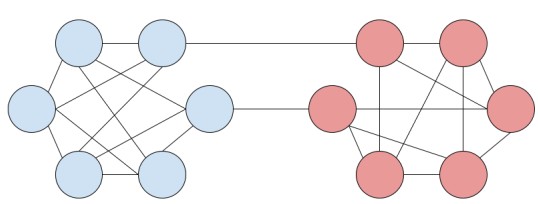

(a) An example of a small expander graph. Any labelling of its nodes cannot exhibit homophily.

(b) Example of a small barbell graph. If a labelling is exactly differentiated between the two groups, then it exhibits homophily.

## 2 BACKGROUND INFORMATION

A graph $G$ is an abstract mathematical model for pairwise relationships, with a set of *vertices* $V$ and a set of *edges* $E \subseteq V \times V$. Two vertices $v_1, v_2$ are said to be connected if $(v_1, v_2) \in E$. For a given graph $G \in \mathcal{G}$ we can also denote its vertices by $V(G)$ and edges $E(G)$. Unless otherwise specified, we assume graphs are undirected and without multi-edges. In machine learning, a graph (or graph-structured data) typically come with a set of node features. Common graph based machine learning tasks include *node classification* (or regression) and *graph classification* (or regression). We use the following notation.

- Graph data $\{G_i = (V_i, E_i)\}_{i=1}^m \in \mathcal{G}$, where $\mathcal{G}$ is the set of all graphs. The neighborhood of a vertex $v$ is denoted $\mathcal{N}(v) = \{u \in V(G_i) : (v, u) \in E(G_i)\}$.
- Node feature $\boldsymbol{x}_v : \mathcal{V} \to \mathcal{X}$, with $\mathcal{X}$ being the feature space, *e.g.*, $\mathcal{X} = \mathbb{R}^{d_v}$.
- Node labels $y : \mathcal{V} \to \mathcal{Y}$, with $\mathcal{Y}$ being the set of labels, *e.g.*, $\mathcal{Y} = [n]$.

**Graph neural networks (GNNs).** GNNs generalize regular neural networks to process data with varying structures and dependencies. GNNs achieve this flexibility via a message passing computational process. In particular, at the $k$-th step (or layer) of message passing, we update the representation $\mathbf{h}_u^{(k+1)}$ of node $u$ as follows,

$$\mathbf{h}_u^{(k+1)} = \text{UPDATE}(\mathbf{h}_u^{(k)}, \text{AGGREGATE}(\{\mathbf{h}_v^{(k)} | v \in \mathcal{N}(u)\})). \tag{1}$$

This update happens for all nodes in parallel within each message passing step. Moreover, the UPDATE and AGGREGATE operators are shared by all nodes, which enables the same GNN to process varying-sized graphs. Once we have finished the finite-step message passing process, we can use the output node representations to make predictions on nodes, edges, and the graph via additionally parameterized readout functions. This message passing framework is quite general since one can instantiate the UPDATE and AGGREGATE operators by different neural networks. For example, the widely used *Graph Convolutional Networks* (GCNs) (Kipf & Welling, 2016), which are the main interest of our work, have the form

$$\mathbf{h}_u^{(k+1)} = \sigma \left( W_k \sum_{v \in \mathcal{N}(u) \cup \{u\}} \frac{\mathbf{h}_v^{(k)}}{\sqrt{|\mathcal{N}(u)|}\sqrt{|\mathcal{N}(v)|}} \right) \tag{2}$$

where one applies a linear transformation ($W_k$) to all node representations, a weighted-sum over the neighborhood, and an element-wise nonlinearity (e.g., ReLU activation). Note that the learnable weights $W_k$ are different from layer to layer.

**Homophily.** A concept studied in network science, homophily (McPherson et al., 2001) is the property that similar nodes group together. For node classification (or node labelling), this means that neighbouring nodes tend to have the same label. Size generalization is plausible when the labelling of the nodes exhibits homophily. The presence of a homophilic graph labelling implies that the labels of the nodes are unlikely to change during the course of a long random walk on the graph.

It is important to note that homophily is also a concept that relates to the graph topology, as not every possible graph structure can be given a labelling that exhibits homophilic properties. An example of one such topology where homophily is impossible is an expander graph (Hoory et al., 2006), as shown in Figure 1a, where nodes have either random or random-like edges connected to a constant number of other nodes in the entire graph. In this case, any labelling of the nodes is far from homophilic, as can be shown using the expansion property. A setting with more homophily is akin to a barbell graph, as shown in Figure 1b, where there are two densely connected components, and comparatively few edges connecting the two dense regions. If the graph labelling of interest lines up with these divisions inherent in the topology, then it is natural to say that it exhibits a homophilic property.

**Cheeger's Inequality.** A mathematical description of homophily can be given using concepts from spectral graph theory. Cheeger's inequality (Hoory et al., 2006) is a theorem that pertains to partitions of graphs, or equivalently binary-valued labellings on graphs (one side of the partition is labelled 0, the other 1). A crucial definition is the *conductance*, defined by

$$\phi(S) = \frac{|E(S, \bar{S})|}{|S|} \quad \forall S \subseteq V \qquad \text{and} \qquad \phi(G) = \min_{|S| \leq \frac{|V|}{2}} \phi(S).$$

Here $E(S, \bar{S})$ is the set of edges connecting a node in $S$ to a node outside of $S$. Cheeger's inequality states

$$\lambda_2/2 \ \leq \ \phi(G) \ \leq \ \sqrt{2\lambda_2},$$

where $\lambda_2$ is the second-smallest eigenvalue of the normalized Laplacian[1] $\tilde{L}$. This inequality links the real-valued quantity $\lambda_2$ to the concept of homophily. If $\lambda_2$ is small then the conductance of $G$ must also be low, by Cheeger's inequality. If a labelling on graph nodes $f : V(\mathcal{G}) \rightarrow \{0, 1\}$ roughly agrees with a low-conductance partition (i.e., one side of the partition $S$ is generally labelled 0 and the complement $\bar{S}$ is generally labelled 1) then the labelling $f$ exhibits homophily.

## 3 IMPROVEMENT OF IN-DISTRIBUTION PAC-BAYES BOUND

The state-of-the-art generalization bounds for GNNs in the in-distribution case were formulated by Liao et al. (2020) using the PAC-Bayes theory. Specifically, they build upon the PAC-Bayes theorem in (Neyshabur et al., 2018) that pertains to homogeneous feedforward neural networks.

We denote one sample as $z = (X, A, y)$ where $X \in \mathcal{X}$, $A \in \mathcal{G}$, and $y \in \mathcal{Y}$ are the node features, the adjacency matrix, and the graph label respectively. Each sample is drawn from some unknown data distribution $\mathcal{D}$ (with support $\mathcal{X} \times \mathcal{G} \times \mathcal{Y}$) in an *i.i.d.* fashion. Since both training and testing samples are drawn from the same distribution, this is the in-distribution setup. Following (Liao et al., 2020), we consider a margin loss for multi-class graph classifications as below,

$$L_{\mathcal{D},\gamma} = L_{\mathcal{D},\gamma}(f_w) = \mathbb{P}_{z\sim\mathcal{D}} \left( f_w(X, A)[y] \leq \gamma + \max_{j\neq y} f_w(X, A)[j] \right) \tag{3}$$

where $\gamma > 0$ is the margin parameter and $f_w$ is the model (hypothesis) parameterized by weights $w$. Since $\mathcal{D}$ is unknown, we can not compute this true loss (risk). We instead minimize the empirical loss (risk) that is defined on the sampled training set $\mathcal{S}$ as below,

$$L_{\mathcal{S},\gamma} = L_{\mathcal{S},\gamma}(f_w) = \frac{1}{m} \sum_{z\in S} \mathbf{1} \left( f_w(X_i, A_i)[y] \leq \gamma + \max_{j\neq y} f_w(X_i, A_i)[j] \right), \tag{4}$$

---

[1]Here $\tilde{L} = D^{-1/2}(D - A)D^{-1/2}$, where $D$ is the diagonal matrix of vertex degrees and $A$ is the adjacency matrix.

where $m$ is the number of training samples. For simplicity, we abbreviate $L_{\mathcal{D},\gamma}(f_w)$ and $L_{\mathcal{S},\gamma}(f_w)$ as $L_{\mathcal{D},\gamma}$ and $L_{\mathcal{S},\gamma}$ respectively from now on.

Our main in-distribution result bounds the gap between true and empirical risks for GCNs, shown in the following theorem. The proof is in Appendix A.1.

**Theorem 3.1.** *For any $B > 0$, $l > 1$, let $f_w \in \mathcal{H} : \mathcal{X} \times \mathcal{G} \to \mathbb{R}^k$ be an l-layer GCN. Then with probability $\geq 1 - \delta$ over the choice of an iid size-m training set $\mathcal{S}$ from the data distribution $\mathcal{D}$, we have for any $w$:*

$$L_{\mathcal{D},0} \leq L_{\mathcal{S},\gamma} + \mathcal{O}\left( \sqrt{\frac{B^2 \boxed{d} l^2 \boxed{(h + \ln l)} \prod_{i=1}^{l} \|W_i\|_2^2 \sum_{i=1}^{l} (\|W_i\|_F^2 / \|W_i\|_2^2) + \ln \frac{m}{\delta}}{\gamma^2 m}} \right) \quad (5)$$

Here $d$ equals to one plus the maximum node degree that can be achieved by the data distribution. $l$ is the depth, *i.e.*, the number of layers, of GCNs. $W_i$ is the weight matrix of GCNs in the $i$-th layer. $B$ is the radius of the minimal $\ell_2$ ball that contains all node features, *i.e.*, $\forall v, \|\boldsymbol{x}_v\|_2 \leq B$. This improves the bound in (Liao et al., 2020), which is provided below for a better comparison,

$$L_{\mathcal{D},0} \leq L_{\mathcal{S},\gamma} + \mathcal{O}\left( \sqrt{\frac{B^2 \boxed{d^{l-1}} l^2 h \boxed{\log(lh)} \prod_{i=1}^{l} \|W_i\|_2^2 \sum_{i=1}^{l}(\|W_i\|_F^2 / \|W_i\|_2^2) + \log \frac{ml}{\delta}}{\gamma^2 m}} \right). \quad (6)$$

The proof of the theorem from (Liao et al., 2020) is an induction over the $l$ layers, in which the spectral norm of the weights and a maximum degree term is multiplied at each step. We observe that it is possible to avoid passing the maximum degree term via a refined argument. This leads to a tightening of one of the main inequalities used in the induction proof, thus in turn resulting in substantial improvements to the overall bound. As can be seen above, we reduce the exponential term $d^{l-1}$ to a linear term $d$, which is a significant improvement for graphs even with small node degrees.

# 4 TOWARDS DEVELOPING A THEORY FOR SIZE GENERALIZATON

In this section, we develop an out-of-distribution (OOD) generalization theory for GNNs. Since we adopt a statistical learning viewpoint, there must necessarily be some assumptions relating the training and testing graphs (otherwise the No-Free Lunch theorem applies). There is a tradeoff between assumptions that are practically relevant, and those for which rigorous guarantees are provable. We have chosen assumptions that we believe strike a balance between those objectives, at least for applications like social networks.

**Size Generalization Assumptions.** We consider the following setup. First, we assume that there exists an extremely large graph $G$ like the user network in Twitter so that one needs to sample subgraphs (*e.g.*, via random walks) for training and testing machine learning models. This is akin to the practical setups of (Grover & Leskovec, 2016; Hamilton et al., 2017). To generate training and testing subgraphs, we run random walks of length $N$ and $M$ respectively on this single large graph, where $M \gg N$, and collect the subgraphs induced by these walks. GNNs are then trained on the subgraphs induced by the shorter (length-$N$) walks. In testing, we assume a procedure where a length-$M$ random walk induced subgraph is sampled from the large subgraph. Random walks are initiated by choosing an initial node uniformly at random from all the nodes in the graph, and at each step there is an equal probability of selecting any of the current node's neighbors. This is an interesting OOD problem where training and testing graphs come from different distributions determined by the underlying large graph and the random walk sampling with specific length. We consider the graph classification problem and assume that the graph label is determined by the majority of node labels within the graph, which is reasonable for many applications that involve homophilic graphs. For the node labeling, we assume it is binary but have no assumptions on how labels are generated.

Crucially, we assume *nothing* about the underlying large graph. Therefore, our setup has advantages over some OOD setups in the literature where a generative model of graphs and labels is explicitly assumed.

**Relation with In-Distribution Result.** We know the relationship between true error defined on the unknown data distribution $\mathcal{D}$ and empirical error defined on the size-$m$ training set $\mathcal{S}$. Specifically, for any GCN $f$, with probability at least $1 - \delta$, we have a general bound as follows,

$$L_{\mathcal{D},0} \leq L_{\mathcal{S},\gamma} + \mathcal{A}(f, \delta, m), \tag{7}$$

where we abbreviate the bound as $\mathcal{A}(f, \delta, m)$ and omit specific parameters like maximum node degree $d$.

In the size generalization problem, we use random walks with lengths $N$ and $M$ for collecting training and testing subgraphs (data) respectively. We are interested in proving a statement of the following form: for any GCN $f$, we have with probability at least $1 - \delta$,

$$L_{\mathcal{D}_M,0} \leq L_{\mathcal{S}_N,\gamma} + \mathcal{B}(f, \delta, m, M, N). \tag{8}$$

The key detail is that $\mathcal{D}_M$ is the distribution of subgraphs induced by random walks **with length $M$** and $\mathcal{S}_N$ is the training set of subgraphs induced by random walks **with length $N$**. Comparing these two losses is the essence of our OOD result. The final term $\mathcal{B}(f, \delta, m, M, N)$ is a general bound involving these parameters.

Based on the in-distribution result like in Theorem 3.1, we can similarly obtain,

$$L_{\mathcal{D}_N,0} \leq L_{\mathcal{S}_N,\gamma} + \mathcal{A}_N(f, \delta, m), \tag{9}$$

where $\mathcal{D}_N$ is the distribution of subgraphs induced by random walks with length $N$ and $\mathcal{A}_N$ is the general bound. The key question boils down to: what is the relationship between $L_{\mathcal{D}_N,0}$ to $L_{\mathcal{D}_M,0}$? This question will be answered in the following sections.

## 4.1 A PROBABILITY BOUND FOR PARTITION CROSSES

The above size generalization problem involves the distributions of random-walk-induced subgraphs from a large graph $G$ with two lengths: $N$ for training and $M$ for testing. Also, $M$ is much larger than $N$. Before we state our results, we would like to explain the simple intuition that motivates our theory:

*If the random walk always stays within the same partition, then the graph label of the random-walk-induced subgraph can be well predicted, no matter how long the random walk is.*

Here a partition means the subset of nodes with the same node label. The goal of this section is to find bounds on $M$ for which we can provide OOD guarantees. We begin by considering a special labelling.

**Special Node Labeling: Sparsest Cut.** A set $S$ that minimizes $\phi(S)$ (and has $|S| \leq |V|/2$) is called a *sparsest cut*. For simplicity assume that $S$ is unique. Using Cheeger's inequality, we first prove the following probability bounds related to this sampling procedure, thereby identifying the length $M$ for which a random walk is likely to stay within the sparsest cut for $d$-regular graphs. The theorems are as follows.

**Theorem 4.1.** *Let $U_M = [u_1, u_2, \ldots, u_M]$ be a length-$M$ random walk over a connected, $d$-regular graph $G$, with $u_1$ chosen from the stationary distribution of the nodes of $G$. If $M \leq d/(2^{5/2}\sqrt{\lambda_2})$, then the probability that $U_M$ crosses the sparsest-cut partition at least once is under $1/2$.*

Here crossing the sparsest-cut partition $S$ means that there exists an edge $(u, v)$ of the random walk satisfies $u \in S$ and $v \in \bar{S}$. $\lambda_2$ is the second-smallest eigenvalue of the normalized Laplacian. We can easily generalize the previous theorem to an arbitrary probability $\delta > 0$ as below.

**Corollary 4.1.1.** *If $M \leq (\delta d)/2^{3/2}\sqrt{\lambda_2}$, the probability of the above random walk $U_M$ crossing over the sparsest-cut partition at least once is at most $\delta$.*

**General Node Labeling.** Theorem 4.1 is restrictive in that it requires the partition $S$ to be the sparsest cut. We now modify the proof to yield a quantity that can work for any node labelling $f$. Specifically, let $\varphi$ be any boolean (*i.e.*, $\{0,1\}$-valued) labelling on the vertices of the graph. Let the *positive node labelling* of $\varphi$ be $S = \{v \in V(G) : \varphi(v) = 1\}$. We are interested in bounding the probability that a random walk of length $M$ includes an edge that crosses the positive node labelling $S$, *i.e.*, an edge $(u, v)$ satisfies $u \in S$ and $v \in \bar{S}$.

**Theorem 4.2.** *Let $\varphi$ be a boolean labelling on the nodes of a connected, $d$-regular graph $G$ with positive node labelling $S$ (0-1 valued vector with $\varphi[i] = 1$ if $v_i \in S$). Let $U_M = [u_1, u_2, \ldots, u_M]$ be a length-$M$ random walk over $G$, with $u_1$ chosen from the stationary distribution of the nodes of $G$. Let $X_i$ be the indicator variable of the event that the $i$-th edge of $U_M$ crosses $S$, i.e., $X_i = \mathbf{1}\left[u_i \in S, u_{i+1} \in \bar{S}\right]$ and $Y_k = \sum_{i=1}^{k} X_i$ is the number of times that $U_M$ crosses $S$ in the first $k$ steps. Let $\varphi' = \varphi - \mathbf{1}(|S|/|V|)$ and $\alpha = \varphi'^{\top} L \varphi' / \|\varphi'\|_2^2$. The conclusion is that:*

$$if \qquad M \leq \frac{d}{2^{5/2}\sqrt{\alpha}} \qquad then \qquad Pr\left[Y_M \geq 1\right] \leq \frac{1}{2}.$$

**Corollary 4.2.1.** *If $M \leq (\delta d)/2^{3/2}\sqrt{\alpha}$, the probability of the above random walk $U_M$ at least crosses over the positive node labelling of $f$ once is at most $\delta$, i.e., $Pr\left[Y_M \geq 1\right] \leq \delta$.*

The formula for $\alpha$ arises from an alternative formulation of Cheeger's inequality which expresses $\lambda_2$ using a Rayleigh quotient (Spielman, 2015), in which $y$ may be viewed as a real-valued labelling on the vertices.

$$\lambda_2 = \min_{y \perp d}(y^{\top} L y)/(y^{\top} D y)$$

## 4.2 SIZE GENERALIZATION ERROR

Recall that, in the size generalization setup, we first train a GNN model $f$ on subgraphs induced by many length-$N$ random walks on $G$. Then during testing, given a large testing subgraph $G_M$ induced by a length-$M$ random walk on $G$, we sample a subgraph $G_N$ via a length-$N$ random walk on $G_M$ and feed it to $f$ to compute the empirical (classification) error for $G_M$. If all nodes of $G_M$ are within a single positive node labelling, then all of their labels are the same. Therefore, no matter which subgraph $G_N$ is sampled, the generalization error (*i.e.*, the probability of making a wrong prediction) for $G_M$ should be the same as the one for $G_N$. Based on this reasoning, we have the following result.

**Theorem 4.3** (Size Generalization Error)**.** *For any $\delta \in [0, 1)$, if we restrict $M$, the size of the large random walk-induced subgraph, such that $M \leq (\delta d)/2^{3/2}\sqrt{\alpha}$, then the in-distribution generalization error $L_{D_M,0}$, i.e., the probability of a wrong prediction on length-$M$-random-walk induced subgraphs, satisfies*

$$L_{\mathcal{D}_M,0} \leq \delta + L_{\mathcal{D}_N,0}. \tag{10}$$

*where $L_{\mathcal{D}_N,0}$ is the in-distribution generalization error of $f$ on length-$N$ random-walk-induced subgraphs.*

Note that this theorem explicitly constrains $M$, whereas the only condition on $N$ is that $L_{\mathcal{D}_N,0}$ is small.

*Proof.* Observe that, for any events $F$ and $E$, we have $\Pr\left[F\right] \leq \Pr\left[E\right] + \Pr\left[F|\bar{E}\right]$. Let $E$ be the event that a length-$M$ random walk crosses the positive node labelling of the ground truth labels, and let $F$ be the event that we make a wrong prediction on the induced subgraph $G_M$. Theorem 3.1 bounds the second term, $\Pr\left[F|\bar{E}\right]$, because the generalization error on $G_M$ is the same as the one on $G_N$ (subgraphs induced by length-$N$ random walks) when $G_M$ does not cross the positive node labelling. Corollary 4.2.1 bounds the first term. Substituting the values from the previous two theorems yields the claimed inequality. $\square$

We already know the bound of the in-distribution generalization error $L_{\mathcal{D}_N,0}$ due to Theorem 3.1 — let us call this quantity $\hat{\delta}$. Using this we can obtain the final result for GCNs under our OOD setup. Theorem 4.3 simply states that, if the length $M \leq (\delta d)/2^{3/2}\sqrt{\alpha}$, with probability at least $1 - \hat{\delta}$, the OOD generalization error on large subgraphs (induced by length-M random walks) is the sum of error $\delta$ and the in-distribution generalization bound on small subgraphs (induced by length-$N$ random walks).

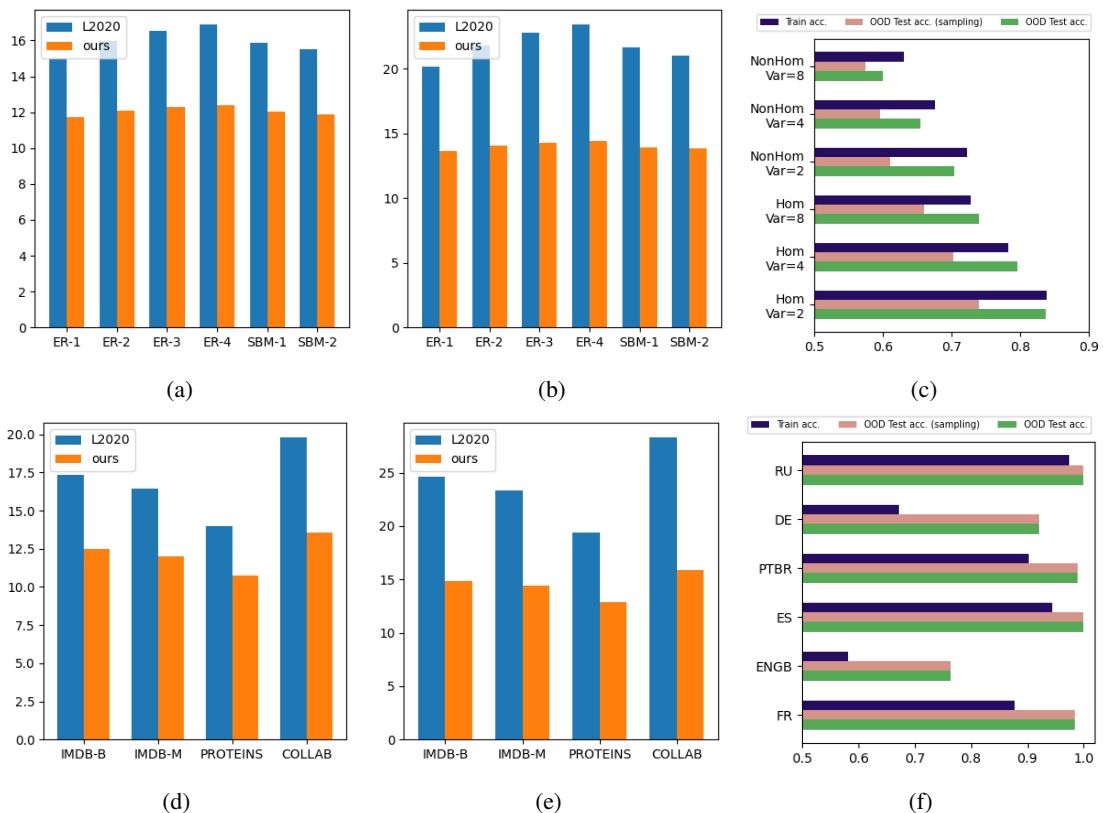

Figure 2: Log-generalization gap values for in-distribution experiments: (a) $l = 4$, Synthetic (b) $l = 6$, Synthetic (d) $l = 4$, Real-world (e) $l = 6$, Real-world. Accuracies for OOD experiments: (c) Synthetic (including both homophilic (Homo) and non-homophilic (NonHomo) graphs) (f) Real-world (Twitch data).

## 5 EXPERIMENTS

### 5.1 IN-DISTRIBUTION: NUMERICAL PAC-BAYES BOUND COMPUTATION

We conduct multi-class graph classification experiments to compare our improved bound to the original PAC-Bayes bound in (Liao et al., 2020). We use the same GCN model, adopt the same datasets, *i.e.*, 6 synthetic datasets obtained from random graph models and 3 real world graph datasets used in (Yanardag & Vishwanathan, 2015), and follow the same experimental protocol. After training a GCN on each dataset, we compute the theoretical bounds using final model. The numerical comparisons of **log bound values** are shown in Figure 2. It is clear that our new bounds are significantly tighter and reduce the bound values by several orders of magnitude. The gap is further increased as the depth increases. The tables of bound values and the specific equations to compute them are provided in Appendix B.1.

### 5.2 OUT-OF-DISTRIBUTION: EFFICACY OF SIZE GENERALIZATION

We performed OOD experiments to validate the values of the upper bound on the size of large subgraphs $M$ that was set in Theorem 4.1 and its related theorems, for synthetic graphs. We also performed experiments on synthetic graphs that were non-homophilic with the same values of $M$ and $N$, to examine size generalization

in this case. We also examined the general feasibility of size generalization in real-world social network data. For synthetic graphs, we calculated this theoretical value for the upper bound, and selected large subgraph size $M$ and small subgraph size $N \ll M$ accordingly. For the real-world case, we chose constant values of $N = 10$ and $M = 50$. For each subgraph, we assign as its graph label the label observed most often among its nodes. After sampling datasets of subgraphs of sizes $M$ and $N$, we train GCN models on the dataset with $N$-length random walks and measure their performance on the training set, the validation set (a smaller data set generated the same way as the train set), and the testing set (a set of subgraphs inuced by length-$M$ random walks). On the test set we record both the performance when inputting the whole large subgraph (Test error), as well as when performing the sampling procedure used for Theorem 4.3, in which we sample an induced subgraph from an $N$-length random walk for each data item (Sampling-test error).

**Synthetic Graphs.** We adopt the CSBMs (Deshpande et al., 2018) to generate graphs that exhibit the homophily property. We use two blocks with much higher probability of connections inside the same block than between blocks, which leads to barbell-like graphs. In the non-homophilic case, we set these probabilities to be equal. We generate binary node labellings via the sparsest cut. CSBMs generate node features via a Gaussian mixture where individual choices of the component are determined by the node label.

**Real-world Graphs.** We used social network data for Twitch streamers from (Rozemberczki et al., 2019). Each node is a streamer (Twitch user), and nodes are connected to mutual friendships. Node features are 3,169 different binary indicators of a wide array of attributes, including games liked, location, etc. Each node is labelled with a boolean value of whether the livestreamer has indicated that they use explicit language.

In all cases, the GCN model achieves OOD test accuracy on large-subgraph that was comparable to ID accuracy on small-subgraph if not outright better. This is even the case when some of the constraints are violated: no $d$-regularity constraint was imposed for any of the datasets, and performance was still good for the test error which did not involve further subgraph sampling. This indicates that the theory is promising in practice for more general forms of size generalization. The accuracy on the train set, test set with subgraph sampling, and unaltered test set are shown in Figure 2, and the numerical values are in Appendix B.2.

For many cases including all real-world cases, the test accuracy was actually higher than the training accuracy. This could potentially indicate that in the cases where size generalization can be guaranteed to work well, the GCN model benefits significantly from extra node information. It is also possible that because of the sampling procedure, there is overlap in nodes between the training and test sets, since they come from random-walk sampling procedures that naively select a uniformly random node as the initial node.

## 6 DISCUSSION

In this work we have expanded the theoretical understanding of the generalizations of GNNs in both in-distribution and out-of-distribution settings, deriving new theoretical guarantees in each setting. The results for in-distribution learning improve upon the state-of-the art PAC-Bayes bounds in (Liao et al., 2020), and the results for out-of-distribution learning provide insight into a practical learning setting under which GNNs are guaranteed to perform effective size generalization. Future directions for the in-distribution understanding would involve lowering the dependencies of other variables like the spectral norm of weights. Generalizing the results to other problems like node classification would also be interesting. In the out-of-distribution case, a number of different observations in experimentation indicate that the theory can still be very much expanded. We have identified cases in real-world datasets where well beyond the bounds on size set forth in the theory, and in all experiments the $d$-regularity assumption is violated, yet GCN size generalization is still effective in these cases. Expansions to the theory, including generalizing to non-$d$-regular graphs, can be explored to explain cases like these.

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

# A MATHEMATICAL PROOFS

## A.1 PROOF OF THEOREM 3.1

The proof is as follows, and makes up the remainder of the chapter.

### A.1.1 IMPROVEMENT ON DEGREE DEPENDENCY

In (Liao et al., 2020), a generalization bound is attained on graph convolutional networks; this bound is dependent on a bound on the maximum perturbation of the function value when a perturbation $U$ is applied to the weights $W$, presented in that paper's Lemma 3.1. The bound is as follows

$$|f_{w+u}(X, A) - f_w(X, A)|_2 \leq eBd^{\frac{l-1}{2}} \left( \prod_{i=1}^{l} \|W_i\|_2 \right) \sum_{k=1}^{l} \frac{\|U_k\|_2}{\|W_k\|_2} \tag{11}$$

The primary goal of this set of improvements is to reduce the factor of $d^{\frac{l-1}{2}}$. For each layer, let $H_i \in \mathbb{R}^{|V| \times h}$ be the matrix containing the hidden embeddings of all of the nodes in its rows, with $h$ being the hidden dimension. In the process of the proof of Theorem 3.1, we are able to show the following:

$$\Phi_j = \max_i |H_j[i, :]|_2 \leq d^{\frac{j}{2}} B \prod_{i=1}^{j} \|W_i\|_2 \tag{12}$$

$$\Psi_j = \max_i |H'_j[i, :] - H_j[i, :]|_2$$

$$\leq Bd^{\frac{j}{2}} \left( \prod_{i=1}^{j} \|W_i\|_2 \right) \sum_{k=1}^{j} \frac{\|U_k\|_2}{\|W_k\|_2} \left( 1 + \frac{1}{l} \right)^{j-k} \tag{13}$$

$$|\Delta_l|_2 = \left| \frac{1}{n} \mathbf{1}_n H'_{l-1}(W_l + U_l) - \frac{1}{n} \mathbf{1}_n H_{l-1} W_l \right|_2$$

$$\leq eBd^{\frac{l-1}{2}} \left( \prod_{i=1}^{l} \|W_i\|_2 \right) \left[ \sum_{k=1}^{l} \frac{\|U_k\|_2}{\|W_k\|_2} \right] \tag{11}$$

We begin to simplify these bounds by removing the dependency on $d^{\frac{j}{2}}$, replacing it instead with a fixed power of $\boxed{d^{1/2}}$ that remains constant for every layer, and thus in the final result of Equation 11 as well.

**Theorem A.1.** *For all $1 \leq j \leq l - 1$, we have:*

$$\Phi_j \leq \boxed{\sqrt{d}} B \prod_{i=1}^{k} \|W_i\|_2 \tag{14}$$

$$\Psi_j \leq \left( 1 + \left( 1 + \frac{1}{l} \right)^j \right) B \boxed{\sqrt{d}} \left( \prod_{i=1}^{j} \|W_i\|_2 \right) \tag{15}$$

*Finally,*

$$|f_{w+u}(X, A) - f_w(X, A)|_2 = |\Delta_l|_2 \leq \left(e + 1 + \frac{2}{l}\right) B \boxed{\sqrt{d}} \prod_{i=1}^{l} \|W_i\|_2 \qquad (16)$$

The proof follows from a lemma about the 2-norm of any node representation at any layer:

**Lemma A.1.1.** *We have, for all $k \in [n]$ and for $j \in [l]$:*

$$|H_j[u, :]|_2 \leq B \boxed{\sqrt{deg(u)}} \left(\prod_{i=1}^{j} \|W_i\|_2\right) \qquad (17)$$

*Proof.* We prove this by induction. By definition $|H_0[u, :]|_2 \leq B$ and thus

$$|H_0[u]| \leq \sqrt{\deg(u)} B \prod_{k=1}^{0} \|W_k\|_2.$$

We assume that for all $u$, we have

$$H_{j-1}[u, :] \leq \sqrt{\deg(u)} B \prod_{k=1}^{j-1} \|W_i\|_2.$$

From these statements we are able to deduce

$$\begin{aligned}
|H_j[u, :]| &\leq \sum_{v \in \mathcal{N}_u} \tilde{L}[u, v] |H_{j-1}[v, :]|_2 \|W_j\|_2 \\
&\leq \sum_{v \in \mathcal{N}_u} \frac{1}{\sqrt{\deg(u)\deg(v)}} \left[\sqrt{\deg(v)} B \prod_{k=1}^{j-1} \|W_k\|_2\right] \|W_j\|_2 \\
&= \sum_{v \in \mathcal{N}_u} \frac{1}{\sqrt{\deg(u)}} B \left(\prod_{k=1}^{j-1} \|W_k\|_2\right) \|W_j\|_2 \\
&= \frac{\deg(u)}{\sqrt{\deg(u)}} B \prod_{k=1}^{j} \|W_k\|_2 \\
&= \sqrt{\deg(u)} B \prod_{k=1}^{j} \|W_k\|_2
\end{aligned}$$

$$(18)$$

$\square$

In these inequalities we use the fact that $\tilde{L}[i, j] = (A + I)_{ij}/\sqrt{\deg(i)\deg(j)}$, and we assume the simple case where there are unweighted edges so that $(A + I)_{ij}$ is 1 if and only if nodes $i$ and $j$ are connected and 0 otherwise.

By Lemma $A.1.1$, we have that $\Phi_j = \max_i |H_j[i, :]|_2 \leq \sqrt{d} B \prod_{i=1}^{j} \|W_i\|_2$, which is exactly the result of equation (14).

**Claim A.1.** *For all $v \in [n]$, $|\Delta_j[v, :]|_2 \leq B\sqrt{deg(v)} \left(1 + \frac{1}{l}\right)^j \left(\prod_{i=1}^{j} \|W_i\|\right) \left(\sum_{i=1}^{j} \frac{\|U_i\|}{\|W_i\|}\right)$*

*Proof.* Proof: We use induction assuming this is true for $\Delta_{j-1}$. We then have

$$|\Delta_j[v, :]|_2 \leq \sum_{u \in \mathcal{N}(v)} \tilde{L}[v, u]|H'_{j-1}[u, :] - H_{j-1}[u, :]|_2 \|W_j + U_j\|_2 + \sum_{u \in \mathcal{N}(v)} \tilde{L}[v, u]|H_{j-1}[u, :]|_2 \|U_j\|_2$$

$$\leq \left[B\left(1 + \frac{1}{l}\right)^{j-1} \left(\prod_{i=1}^{j-1} \|W_i\|\right) \left(\sum_{i=1}^{j-1} \frac{\|U_i\|_2}{\|W_i\|_2}\right) \|W_j + U_j\| + B\|U_j\| \prod_{i=1}^{j-1} \|W_i\|\right] \quad (19)$$

$$\left(\sum_{u \in \mathcal{N}(v)} \tilde{L}[v, u]\sqrt{\deg(u)}\right)$$

$$= B\sqrt{\deg(v)} \prod_{i=1}^{j-1} \|W_i\| \left[\|W_j + U_j\| \left(1 + \frac{1}{l}\right)^{j-1} \left(\sum_{i=1}^{j-1} \frac{\|U_i\|_2}{\|W_i\|_2}\right) + \|U_j\|\right]$$

$$= B\sqrt{\deg(v)} \prod_{i=1}^{j} \|W_i\| \left[\frac{\|W_j + U_j\|_2}{\|W_j\|_2} \left(1 + \frac{1}{l}\right)^{j-1} \left(\sum_{i=1}^{j-1} \frac{\|U_i\|_2}{\|W_i\|_2}\right) + \frac{\|U_j\|_2}{\|W_j\|_2}\right]$$

$$\leq B\sqrt{\deg(v)} \prod_{i=1}^{j} \|W_i\| \left[\left(1 + \frac{1}{l}\right)^{j} \left(\sum_{i=1}^{j-1} \frac{\|U_i\|_2}{\|W_i\|_2}\right) + \frac{\|U_j\|_2}{\|W_j\|_2}\right]$$

$$\leq B\sqrt{\deg(v)} \prod_{i=1}^{j} \|W_i\| \left(1 + \frac{1}{l}\right)^{j} \left(\sum_{i=1}^{j} \frac{\|U_i\|_2}{\|W_i\|_2}\right) \quad (20)$$

$\Delta_l$ has a slightly different formulation but it has a very similar bound:

$$|\Delta_l|_2 = \left|\frac{1}{n}\mathbf{1}_n \left(\tilde{L}H'_{l-1}(W_l + U_l) - \frac{1}{n}\mathbf{1}_n\tilde{L}H_{l-1}(W_l)\right)\right|_2$$

$$= \frac{1}{n}\left|\mathbf{1}_n\tilde{L}(H'_{l-1} - H_{l-1})(W_l + U_l) + \mathbf{1}_n\tilde{L}H_{l-1}(U_l)\right|_2$$

$$\leq \frac{1}{n}\sum_{i=1}^{n} |\Delta_{l-1}[i, :]|_2\|W_l + U_l\|_2 + \frac{1}{n}\sum_{i=1}^{n} |H_{l-1}[i, :]|_2\|U_l\|_2$$

$$\leq B\sqrt{d}\prod_{i=1}^{l-1} \|W_i\| \left(1 + \frac{1}{l}\right)^{l-1} \left(\sum_{i=1}^{l-1} \frac{\|U_i\|_2}{\|W_i\|_2}\right) \|W_l + U_l\|$$

$$+ B\sqrt{d}\|U_l\|_2 \prod_{i=1}^{l-1} \|W_i\|_2$$

$$\leq B\sqrt{d}\prod_{i=1}^{l} \|W_i\| \left[\left(1 + \frac{1}{l}\right)^{l} \left(\sum_{i=1}^{l-1} \frac{\|U_i\|}{\|W_i\|}\right) + \frac{\|U_l\|}{\|W_l\|}\right]$$

$$\leq B\sqrt{d}\prod_{i=1}^{l} \|W_i\| \left(1 + \frac{1}{l}\right)^{l} \left(\sum_{i=1}^{l} \frac{\|U_i\|}{\|W_i\|}\right)$$

$$\leq eB\sqrt{d}\prod_{i=1}^{l}\|W_i\| \left(\sum_{i=1}^{l}\frac{\|U_i\|}{\|W_i\|}\right) \tag{21}$$

$\square$

From this we have proven a tighter bound on the final output of the GNN under perturbation, which we will use to calculate probabilistic and generalization bounds.

### A.1.2 IMPROVEMENT ON PROBABILISTIC BOUNDS USING RANDOM MATRIX THEORY

In (Liao et al., 2020), for all $i \in [l]$, with $l$ being the number of layers, the prior and the distribution of the perturbations $U_i \in \mathbb{R}^{d_{i+1} \times d_i}$,, where all hidden dimensions $d_i$ are upper-bounded by a value $h$, were generated by a normal distribution $\mathcal{N}(0, \sigma^2 I)$, and give probabilistic bounds on the operator norms $\|U_i\|$ as $P(\forall i, \|U_i\| \leq t)$ with probability greater than $1 - 2lh \exp -t^2/2h\sigma^2$. We improve these bounds using theorems on random matrices from work on high-dimensional probability, namely (Vershynin, 2018).

**Theorem A.2** (Theorem 4.4.5 in (Vershynin, 2018)). *Let $A$ be a matrix in $\mathbb{R}^{m \times n}$, where the entries $A_{ij}$ are independent, mean-zero, sub-Gaussian random variables. Then, for all $t > 0$ we have*

$$\|A\| \leq CK(\sqrt{m} + \sqrt{n} + t)$$

*with probability $\geq 1 - \exp(-t^2)$, where $K = \max_{i,j}\|A_{ij}\|_{\psi_2}$ and $C$ is some constant.*

In the above theorem the norm $\|X\|_{\psi_2}$ is defined as $\inf\{t : \mathbb{E}[\exp(X^2/t^2)] \leq 2\}$. In Example 2.5.8 in ($Vershynin$, 2018), it is shown that if $X \sim \mathcal{N}(0, \sigma^2)$ then it has $\|X\|_{\psi_2} \leq C\sigma$.

**Corollary A.2.1.** *If $U \in \mathbb{R}^{m \times n}$ is a random matrix generated with the distribution $\mathcal{N}(0, \sigma^2 I)$ (i.e. all entries are independent and identically distributed Gaussian random variables), then we have*

$$\|U\| \leq \sigma(\sqrt{m} + \sqrt{n} + t)$$

*with probability at least $1 - 2\exp(-t^2)$.*

With a change of variable, we are able to calculate the following:

$$P(\forall i.\|U_i\|_2 \leq t) \geq 1 - P(\exists i, \|U_i\| > t)$$

$$\geq 1 - \sum_{i=1}^{l} P(\|U_i\| > t)$$

$$\geq 1 - 2l\exp\left(\left(\frac{t}{C\sigma} - 2\sqrt{h}\right)^2\right)$$

And by setting the right-hand side to 1/2, we obtain:

$$t = C\sigma(2\sqrt{h} + \sqrt{\ln(4l)})$$

Using the above equation combined with our bound we are able to get

$$
\begin{aligned}
|f_{w+u}(X,A) - f_w(X,A)|_2 &\leq eB\sqrt{d}l \left( \prod_{i=1}^{l} \|W_i\|_2 \right) \sum_{k=1}^{l} \frac{\|U_k\|_2}{\|W_k\|_2} \\
&= eB\sqrt{d}\beta^l l \sum_{k=1}^{l} \frac{\|U_k\|_2}{\beta} \\
&\leq eB\sqrt{d}\beta^{l-1}l(\sigma(2\sqrt{h} + \sqrt{\ln(4l)})) \\
&\leq e^2 B\sqrt{d}\tilde{\beta}^{l-1}(\sigma(2\sqrt{h} + \sqrt{\ln(4l)})) \leq \frac{\gamma}{4}
\end{aligned}
$$
(22)

Here $\tilde{\beta}$ is an estimated of $\beta$ such that $|\beta - \tilde{\beta}| \leq \beta/l$ that can be generated *a priori*; we discuss this in a later subsection.

We can set $\sigma = \frac{\gamma}{4e^2 B\tilde{\beta}\sqrt{d}C\left(2\sqrt{h}+\sqrt{\ln(4l)}\right)}$ to satisfy the final inequality. From this we can calculate the KL-divergence between the posterior and the prior:

$$
\begin{aligned}
\mathrm{KL}(Q\|P) = \frac{|w|_2^2}{2\sigma^2} &= \frac{16e^4 B^2 dl^2 \beta^{2(l-1)}\left(2\sqrt{h}+\sqrt{\ln(4l)}\right)^2}{2\gamma^2} \sum_{i=1}^{l} \|W_i\|_F \\
&\leq \mathcal{O}\left( \frac{B^2 d\beta^{2l}l^2(h+\ln(l))}{\gamma^2} \sum_{i=1}^{l} \frac{\|W_i\|_F^2}{\beta^2} \right) \\
&\leq \mathcal{O}\left( B^2 dl^2 \boxed{(h+\ln(l))} \frac{\prod_{i=1}^{l}\|W_i\|^2}{\gamma^2} \sum_{i=1}^{l} \frac{\|W_i\|_F^2}{\|W_i\|^2} \right)
\end{aligned}
$$
(23)

From this we are able to calculate the generalization bound and thus prove the theorem.

$$
L_{\mathcal{D},0} \leq L_{\mathcal{S},\gamma} + \mathcal{O}\left( \sqrt{\frac{B^2 dl^2(h+\ln(l))\prod_{i=1}^{l}\|W_i\|_2^2 \sum_{i=1}^{l}\frac{\|W_i\|_F^2}{\|W_i\|_2^2} + \ln\frac{m}{\delta}}{\gamma^2 m}} \right)
$$
(24)

### A.1.3 SELECTING PARAMETER $\tilde{\beta}$

The prior normal distribution's variance parameter $\sigma^2$ is dependent on $\beta$, but $\beta$ cannot be used in its calculation because that information is only known after model training. Instead, we can select a parameter $\hat{\beta}$ such that $|\beta - \hat{\beta}| \leq \frac{1}{l}\beta$ and thus $\frac{1}{e}\beta^{l-1} \leq \hat{\beta}^{l-1} \leq e\beta^{l-1}$ (as per equation 33 in (Liao et al., 2020)).

As in (Liao et al., 2020) we only have to consider values of $\beta$ in the range

$$
\left( \frac{\gamma}{2B\sqrt{d}} \right)^{1/l} \leq \beta \leq \left( \frac{\gamma\sqrt{m}}{2B\sqrt{d}} \right)^{1/l}
$$

as otherwise the generalization bound holds trivially because $L_{\mathcal{D},0} \leq 1$ by definition.

If we consider values of $\hat{\beta}$ that cover this interval then by union bound we are still able to get a high probability; the covering $C$ needs to have $|C| = \frac{l}{2}(m^{\frac{1}{2l}} - 1)$.

## A.2 PROOFS OF OUT-OF-DISTRIBUTION PROBABILITY BOUNDS

### A.2.1 PROOF OF THEOREM 4.1

*Proof.* Because $u_1$ is chosen from the stationary distribution (uniform over vertices, because G is connected and $d$-regular), then for all $i \geq 1$ the distribution for $u_i, u_{i+1}$ follows the distribution Unif$[E]$, where $E$ is the edge set of the graph.

Let $S$ be the sparsest-cut partition of $G$. Let $X_i$ be the indicator of the event that the vertex pair is in the set of edges crossing the partition, namely $\mathbf{1}\{(u_i, u_{i+1}) \in E(S, \bar{S})\}$. By linearity of expectation, this means that $E[X_i] = |E(S, \bar{S})|/|E|$.

Furthermore, let $Y_k$ be the cumulative number of edges crossing the partition along the first $k$ steps of the random walk. This is expressed nicely as $Y_k = \sum_{i=1}^{k} X_i$. Thus $E[Y_k] = k\frac{|E(S,\bar{S})|}{|E|}$.

Applying Markov's inequality, we get $\Pr[Y_k \geq tk|E(S, \bar{S})|/|E|] \leq 1/t$. Suppose we wish to examine under what conditions we can ensure that we do not cross over the partition at all in $M$ steps, i.e. $\Pr[Y_M \geq 1] \leq 1/2$. From the inequality above, we are able to get that

$$\Pr\left[Y_M \geq 2M\frac{|E(S,\bar{S})|}{|E|}\right] \leq \frac{1}{2}$$

just by setting $k = M$ and $t = 2$. We then use the following basic fact: if we have an inequality of the form $\Pr[Z \geq z] \leq \frac{1}{2}$, then $\Pr[Z \geq z'] \leq \frac{1}{2}$ for any $z' \geq z$.

Let $E(S)$ denote the set of edges connected to any vertex in $S$. Because $|E(S)| \leq |E|$, then we have $|E(S, \bar{S})|/|E| \leq |E(S, \bar{S})|/|E(S)|$. Furthermore, since we assume a connected graph, $|E(S)| \geq (d/2)|S|$, and thus $|E(S, \bar{S})|/|E(S)| \leq |E(S, \bar{S})|/[(d/2)|S|]$. [2] Thus using the fact above we can deduce

$$\Pr\left[Y_M \geq 2M\frac{|E(S,\bar{S})|}{(d/2)|S|}\right] \leq \frac{1}{2}$$

Note that $|E(S, \bar{S})|/|S|$ is the conductance of the graph $\phi(G)$, because $S$ was defined to be the sparsest-cut partition of $G$. Thus we can apply the fact again with Cheeger's inequality to get

$$\Pr\left[Y_M \geq 2M(2/d)\sqrt{2\lambda_2}\right] \leq \frac{1}{2}$$

And since we are interested in $\Pr[Y_M \geq 1]$, we can thus set $2M\sqrt{2\lambda_2} \leq 1$ to get a necessary condition for $M$, from which we achieve

$$M \leq \frac{d}{2^{5/2}\sqrt{\lambda_2}}$$

This completes the proof. □

---

[2]It is important to note that this specific dependency of $|E(S)|$ on $d$ requires $G$ to be a $d$-regular graph. If the theorem is to be expanded to more general cases, one may use the simple inequality $|E(S)| \geq |S|$.

### A.2.2 PROOF OF THEOREM 4.2

*Proof.* The quantity $\varphi'$ is a transformation of $\varphi$ that retains all the information contained in $\varphi$ while still being orthogonal to the all-ones vector $\mathbf{1}$, so that we can apply Cheeger's inequality. This orthogonalization is rather standard and can be found in (Spielman, 2015).

Let $s = |S|/|V(G)|$. Note that $s \in [0,1]$, and without loss of generality we can assume that $s \leq 1/2$.

We observe that the $v$th coordinate of the vector $\varphi'$ corresponds to the mapping

$$\varphi'(v) = \begin{cases} 1 - s & v \in S \\ -s & v \notin S \end{cases} \tag{25}$$

This ensures that $\varphi'$ is orthogonal to $\mathbf{1}$, as

$$\varphi'^\top \mathbf{1} = \sum_{i=1}^n \varphi'(v_i) = |S| \left(1 - \frac{|S|}{|V|}\right) + (|V| - |S|) \left(-\frac{|S|}{|V|}\right) = |S| - |V| \left(\frac{|S|}{|V|}\right) = 0.$$

We then note that $\|\varphi'\|_2^2 = \sum_{i=1}^n \varphi(v)^2$ is equal to $s(1-s)|V|$, and we can infer $|S|/2 \leq \|\varphi'\|_2^2 \leq |S|$; the first inequality holds since $s \leq 1/2$.

The number of edges $|E(S, \bar{S})|$ crossing the labelling-partition is equal to $\varphi'^\top L \varphi'$, as

$$\varphi'^\top L \varphi' = \sum_{(u,v) \in E} ((\varphi(u) - s) - (\varphi(v) - s))^2 = |E(S, \bar{S})|$$

where $L$ is the Laplacian matrix of $\mathcal{G}$.

Thus the quantity $2M \frac{|E(S,\bar{S})|}{|E(S)|} \leq 2M \frac{\varphi'^\top L \varphi'}{|E(S)|} \leq 2M \frac{\varphi'^\top L \varphi'}{(d/2)|S|}$. We are able to get the second inequality because we know $|E(S)| \geq (d/2)|S|$. Because we know that $|S| \geq \|\varphi'\|^2$, we can then upper bound this further by $2M \frac{\varphi'^\top L \varphi'}{(d/2)\|\varphi'\|_2^2}$. Substituting this quantity in the proof of Theorem 4.1, we achieve the desired bound for $M$. $\qquad\square$

## B EXPERIMENTAL METHODOLOGY AND RESULTS

### B.1 IN-DISTRIBUTION EXPERIMENTS

The datasets used are a combination of synthetic (Erdos-Renyi and Stochastic Block Model) and real-world graphs (IMDBBINARY and IMDBMULTI of data from the Internet Movie Database, and COLLAB, a dataset of academic collaborations), and a bioinformatics dataset, PROTEINS, from (Yanardag & Vishwanathan, 2015).

Two different GCN network depths of of $l = 4$ and $l = 6$ were used.

We use the following formulae for the generalization bound from (Liao et al., 2020) and our new bound, using an explicit constant factor of 42 from (Liao et al., 2020).

$$\text{GenGap}(B, d, l, \{W_i\}_{i=1}^l) = \sqrt{42 \cdot \frac{B^2 d^{l-1} l^2 \ln(4lh) \prod_{i=1}^l \|W_i\|_2^2 \sum_{i=1}^l \frac{\|W_i\|_F^2}{\|W_i\|_2^2}}{\gamma^2 m}} \tag{26}$$

Similarly, the formula used for the new PAC-Bayes generalization bound is

$$\text{GenGap}(B, d, l, \{W_i\}_{i=1}^l) = \sqrt{42 \cdot \frac{B^2 dl^2(h + \ln(l)) \prod_{i=1}^l \|W_i\|_2^2 \sum_{i=1}^l \frac{\|W_i\|_F^2}{\|W_i\|_2^2}}{\gamma^2 m}} \tag{27}$$

We remove an additive $O(\log m)$ term in the numerator within the square root after validating that it was numerically negligible.

Tables below are for calculated bounds in the case of 4 layers (Table 1) and 6 layers (Table 2).

| Dataset | (Liao et al., 2020) | New bound |
|---|---|---|
| ER-1 | $14.985735 \pm 0.071$ | $11.727639 \pm 0.071$ |
| ER-2 | $15.972616 \pm 0.068$ | $12.080796 \pm 0.068$ |
| ER-3 | $16.533779 \pm 0.038$ | $12.285284 \pm 0.038$ |
| ER-4 | $16.869156 \pm 0.000$ | $12.391819 \pm 0.000$ |
| SBM-1 | $15.859210 \pm 0.080$ | $12.030569 \pm 0.080$ |
| SBM-2 | $15.503044 \pm 0.066$ | $11.892126 \pm 0.066$ |
| IMDBBINARY | $17.370839 \pm 0.079$ | $12.458184 \pm 0.079$ |
| IMDBMULTI | $16.466189 \pm 0.038$ | $11.977553 \pm 0.038$ |
| COLLAB | $19.773157 \pm 0.009$ | $13.574678 \pm 0.009$ |
| PROTEINS | $14.011104 \pm 0.079$ | $10.753008 \pm 0.079$ |

Table 1: Table of generalization bounds, 4 layers (log values)

| Dataset | (Liao et al., 2020) | New bound |
|---|---|---|
| ER-1 | $20.123308 \pm 0.046$ | $13.607115 \pm 0.046$ |
| ER-2 | $21.814967 \pm 0.017$ | $14.031327 \pm 0.017$ |
| ER-3 | $22.782537 \pm 0.008$ | $14.285547 \pm 0.008$ |
| ER-4 | $23.377554 \pm 0.021$ | $14.422881 \pm 0.021$ |
| SBM-1 | $21.605130 \pm 0.037$ | $13.947847 \pm 0.037$ |
| SBM-2 | $21.028283 \pm 0.047$ | $13.806447 \pm 0.047$ |
| IMDBBINARY | $24.648394 \pm 0.058$ | $14.823084 \pm 0.058$ |
| IMDBMULTI | $23.373624 \pm 0.105$ | $14.396352 \pm 0.105$ |
| COLLAB | $28.261012 \pm 0.106$ | $15.864055 \pm 0.106$ |
| PROTEINS | $19.421826 \pm 0.059$ | $12.905633 \pm 0.059$ |

Table 2: Table of generalization bounds, 6 layers (log values)

## B.2 OUT-OF-DISTRIBUTION EXPERIMENTS

### B.2.1 METHODOLOGY

Experiments were performed to measure the effectiveness size generalization of GCN models when applied to the size generalization learning case described in Section 4, where the learning task is classifying the most common node label in sub-communities of a large underlying network.

For each of the synthetic graphs, we calculate an upper bound for $M$ set in the out-of-distribution inequalities we have derived. Since the graphs examined are all not $d$-regular, we calculate a value of $\alpha$ as $\frac{\varphi^\top L \varphi}{\varphi^\top D \varphi}$, where $L$ is the graph Laplacian matrix and $D$ is the diagonal degree matrix, to apply to the formula set in Theorem 4.2. Furthermore, we use a more permissive value of $\delta = 0.75$.

Similar upper bounds for $M$ were computed for the real-world cases, but the values were too small for experimental use. In this case, we just set $N = 10$ and $M = 50$ to attempt to gain insight about the size generalization task's general feasibility in real-world cases.

All experiments were performed with use of the Adam optimizer (Kingma & Ba, 2015), with a constant learning rate 0.01. Models were trained for 10 epochs, with a batch size 32 randomly selected.

The models used are different parameterizations of the Graph Convolutional Network as implemented by the library `pytorch-geometric` (Fey & Lenssen, 2019). For synthetic experiments, which used smaller graphs with generally smaller degree, the parameterization was 3 layers with a hidden dimension of 5, and for the real-world data case, the parameterization was 10 layers with a hidden dimension of 32.

For each underlying graph, we generate three train/validation sets (of size $N$ random walks) and test sets (of size $M$ random walks) and we record the loss and accuracy as the average of the three runs.

### B.2.2 SYNTHETIC GRAPH EXPERIMENTS

A large underlying synthetic graph was generated using the stochastic block model, with some adjustment to ensure that the randomly-generated graph had a single connected component. By controlling the intra- and inter-block connection probability values, we are able to control the homophily of the generated graph, which we validate by measuring the value of $\lambda_2$, as well as calculating the sparsest cut via "Cheeger rounding" (Spielman, 2015) and subsequently the conductance of the graph with respect to this partition.

In the experiments, we generated a graph with approximately 2000 nodes, with in-block connectivity probability set to $8/1000$ and inter-block connectivity set to $6/10^5$.

Node features are generated from a mixture of multivariate Gaussian distributions with dimension 3, mean $(-0.5, -0.5, -0.5)$ for one block, and mean $(0.5, 0.5, 0.5)$ for the other; the covariance matrix is a diagonal matrix (each coordinate is independent) of variance either 2, 4, or 8.

Experiments were also performed on non-homophilic synthetic graphs. Like the homophilic synthetic graphs they are generated with the stochastic block model with about 2000 nodes, about 1000 of each label, and with the same mixture-of-Gaussian node features. However the parameters used for the generation of connection are crucially different. The probabilities of connection between nodes of the same block and nodes of a different block are set to be equal, with both being set to $8/1000$. These settings ensure that a node's label is independent from the labels of its neighbors, so the homophily property is not exhibited.

Contrasting with the results shown for the homophilic synthetic graphs, the non-homophilic graph results show that the out-of-distribution test accuracy is less than the training accuracy. This further illustrates the association between homophily and size generalization.

### B.2.3 REAL-WORLD GRAPH EXPERIMENTS

Since the node features are indicators, we encoded the node feature information by using the positional encoding mechanism introduced in the Transformer model (Vaswani et al., 2017). For each node, each of their integer indicators was encoded via positional embedding and aggregated via sum.

| Graphs | $\frac{0.75}{2^{\frac{3}{2}}\sqrt{\alpha}}$ | $N$ | $M$ | Train loss | Train acc. | Val. loss | Val. acc. | Test loss (sampling) | Test acc. (sampling) | Test loss | Test acc. |
|---|---|---|---|---|---|---|---|---|---|---|---|
| CSBM var=2 | 20.7 | 10 | 20 | 0.005 | 0.953 | 0.007 | 0.943 | 0.020 | 0.869 | 0.003 | 0.970 |
| CSBM var=4 | 18.6 | 10 | 18 | 0.014 | 0.889 | 0.014 | 0.897 | 0.026 | 0.811 | 0.009 | 0.937 |
| CSBM var=8 | 17.5 | 10 | 17 | 0.021 | 0.826 | 0.022 | 0.803 | 0.032 | 0.745 | 0.015 | 0.880 |
| CSBM non-homophilic var=2 | 5.12 | 10 | 20 | 0.032 | 0.723 | 0.032 | 0.720 | 0.049 | 0.611 | 0.034 | 0.704 |
| CSBM non-homophilic var=4 | 5.14 | 10 | 18 | 0.036 | 0.676 | 0.041 | 0.643 | 0.049 | 0.596 | 0.038 | 0.655 |
| CSBM non-homophilic var=8 | 5.12 | 10 | 17 | 0.042 | 0.630 | 0.045 | 0.570 | 0.049 | 0.575 | 0.043 | 0.600 |
| Twitch-ENGB | – | 10 | 50 | 0.042 | 0.581 | 0.044 | 0.570 | 0.025 | 0.764 | 0.025 | 0.764 |
| Twitch-FR | – | 10 | 50 | 0.013 | 0.877 | 0.012 | 0.887 | 0.002 | 0.984 | 0.002 | 0.984 |
| Twitch-ES | – | 10 | 50 | 0.006 | 0.944 | 0.008 | 0.923 | 0.000 | 0.999 | 0.000 | 0.999 |
| Twitch-PTBR | – | 10 | 50 | 0.010 | 0.902 | 0.011 | 0.893 | 0.001 | 0.989 | 0.001 | 0.989 |
| Twitch-DE | – | 10 | 50 | 0.035 | 0.672 | 0.032 | 0.690 | 0.008 | 0.921 | 0.008 | 0.921 |
| Twitch-RU | – | 10 | 50 | 0.003 | 0.974 | 0.004 | 0.973 | 0.000 | 1.000 | 0.000 | 1.000 |

Table 3: Experimental data for out-of-distribution experiments

