# OpenReview forum: "In-distribution and Out-of-distribution Generalization for Graph Neural Networks"
_ICLR.cc/2023/Conference — Submitted to ICLR 2023_

### Official Review · Reviewer_KCgB · 2022-10-19

**Confidence:** 3
**Correctness:** 4
**Technical Novelty And Significance:** 3
**Empirical Novelty And Significance:** 2
**Recommendation:** 6

**Clarity, Quality, Novelty And Reproducibility:**

# Clarity:
The intuitions provided in the main paper are clear and the arguments and intuitions are well-explained. In the appendix, the proof of Theorem 3.1 is a bit hard to follow. I regularly had to refer back to different parts of the proof to understand the respective quantities and variables. Moreover, some steps in the proof (namely, the inequality sequences) can be better explained.

# Quality:
The paper's contributions are meaningful and its arguments all appear sound and well-motivated.

# Novelty:
The results and perspective on out-of-distribution generalization are novel.

# Reproducibility:
N/A to proofs. Experimental results appear to be easily reproducible.

**Strength And Weaknesses:**

# Strengths:
- The improvement of the in-distribution bound is significant and meaningful. The proof of this theorem also appears to be sound
- The setting proposed for out-of-distribution analysis is interesting and avoids making assumptions on the inherent graph structure, and thus offers a novel approach to studying how graph convolutional networks can generalize to larger graphs.

# Weaknesses:
- Though the out-of-distribution setup is interesting and avoids assumptions on the structure of the train and test graphs, the assumptions it introduces instead appear to be limiting. In particular, the graph labelling objective seems restrictive and specialized. Typically, graph classification objectives go far beyond node statistics (majority of labels) and explore structural graph properties, which lies outside the scope of this analysis. It is also not at all clear to me how one can move beyond this limitation: The very essence of the probabilistic argument in the paper relies on aggregating over node classes, and thresholding the probability of transitioning between classes in a random walk: Without a node-based objective, such an analysis is not relevant. Therefore, I am afraid this approach cannot be used to develop a more general (objective-agnostic) framework for out-of-distribution generalization. I understand that assumptions inevitably must be made about a connection between the train and test sets (shared structure, or in this case similar walk outcomes with respect to the objective), however I feel that simplifying assumptions on the objective function will not lead to general insights on out-of-distribution learning, as claimed in the paper. I therefore suggest that the authors clarify the limitations and scope of their analysis more extensively.

- On a more minor note, it would be interesting to mention how/if the results extend beyond the GCN architecture. Currently, the result and proof requires a GCN, but it would be useful to provide an intuition as to how such a framework can be applied to other models.

**Summary Of The Paper:**

The paper studies in-distribution and out-of-distribution generalization with graph convolutional networks. In the former setting, the paper improves on an existing bound in the literature by reducing dependency on maximum node degree to a linear factor. In the latter setting, the paper builds on in-distribution results, and considers the case of homophilic (but arbitrary) graph structures. More specifically, the out-of-distribution setup considers a single very large graph from which smaller training graphs and larger testing graphs are sampled using random walks. Then, the paper considers the majority voting problem, in which the binary label of a graph is the binary label of the majority of its constituent nodes. Using a probabilistic argument, the paper then shows that, for homophilic graphs, the probability of conducting a random walk that crosses the label divide, i.e., from label 0 to label 1 or vice versa, in the original large input graph can be thresholded to a quantity delta with reasonable bounds on the test graph size M and training graph size N. This probability can then be considered jointly with the probability error in the in-distribution theorem to yield an overall error probability for this out-of-distribution setting. Finally, the paper conducts an empirical analysis on a variety of synthetic and real-world datasets, showing performance levels consistent with the theory, and even demonstrating strong performance when certain assumptions are violated (degree regularity for example).

**Summary Of The Review:**

All in all, I believe that the paper makes a useful contribution for studying the generalization performance of GCNs. I find the in-distribution result to be strong and potentially impactful. I also appreciate the perspective offered in the out-of-distribution analysis, and how this avoids making assumptions on the graph structure. However, I have concerns about the restriction to the objective function, which in my opinion casts doubts on the general applicability of the approach, particularly its probabilistic argument for scaling the in-distribution theorem to the out-of-distribution setting. All in all, I am learning towards supporting the paper. I am also happy to hear back from the authors about my concerns, and am willing to support the paper more strongly should my concerns be addressed.

---

> ### Author Response · Authors · 2022-11-23
> **Response to Reviewer KCgB**
>
> We thank the reviewer for the positive comments!
>
> > **Q1: Assumptions of OOD generalization and their limitations.**
>
> A1: We added more discussion (e.g., limitations and scope of our analysis) and justification in the introduction and Section 4 in the revision. Please see our general response for more details.
>
> > **Q2: it would be interesting to mention how/if the results extend beyond the GCN architecture**
>
> A2: Thanks for raising this point!
>
> Our proof techniques for the in-distribution results could be generalized to message passing GNNs (MPGNNs) following Liao et al. (2021) since the way to handle geometric series in the perturbation analysis of MPGNNs are similar to the one we used in Theorem A.1. But the overall results and derivation will be quite cumbersome for MPGNNs due to their more complicated architectures. We thus leave it as future work.
>
> The OOD generalization analysis is architecture agnostic so that it could be applied to any GNNs. One just needs to instantiate the corresponding in-distribution generalization error in Theorem 4.3.

---

> > ### Comment · Reviewer_KCgB · 2022-11-29
> > **Reviewer Response**
> >
> > Thank you for your response! I appreciate the additional experiments and the discussion. I think the changes certainly improve the paper and make the contribution better defined. However, I still have concerns about the applicability and generality of the work. As a result, I will keep my current verdict.

---

### Official Review · Reviewer_h6gY · 2022-10-24

**Confidence:** 3
**Correctness:** 3
**Technical Novelty And Significance:** 3
**Empirical Novelty And Significance:** 2
**Recommendation:** 6

**Clarity, Quality, Novelty And Reproducibility:**

I can attest to the clarity and novelty of the work. The paper is written without many assumptions of prior knowledge of GNN generalizability. The improved in-distribution bounds are an original contribution building on recent work in the literature. My major concerns are the size of contribution which are detailed in the main review.

**Strength And Weaknesses:**

Pros:
* The paper tackles an important topic in better understanding the theoretical generalization bounds of GNNs. Further, the paper is comprehensive in looking at both in and out of distribution generalization.
* The paper is clearly written and understandable for a reader familiar with GNN/GCNs but not with theoretical generalization bounds of these models.

Cons:
* For the in-distribution bounds it is not fully clear what is the significance / level of contribution of tightening the bounds in Liao et al. To me this could be better shown with a more detailed analysis of the experimental results in sec. 5.1 in particular for the real-world graphs. As is the log-generalization gap values are difficult to interpret and put into perspective.
* It would be helpful to have more specification for when the expression for M in Theorem 4.3 is compatible with the condition that M >> N.
* In analyzing the size generalization, it can be argued that the paper translates an out-of-distribution question into an in-distribution one. Specifically the paper largely side steps situations in which it is desirable to test the GNN on the large subgraph itself instead of sampling from the test subgraph, for instance the setting considered in “From Local Structures to Size Generalization in Graph Neural Networks”.
* It is helpful that the authors explain the surprising findings in Figures 2c and 2f; however, it would be helpful to also include cases where the test accuracy is not higher than the training accuracy. When the test accuracy is always higher the results preclude the need to understand out-of-distribution generalization.

Minor comments:
* On page five when describing the size generalization setup in the following sentence: “In testing, we assume a procedure where a length-N random walk induced subgraph is sampled from the large subgraph”, it was not clear to me that the large subgraph is the length-M random-walk induced graph until later.

**Summary Of The Paper:**

The objective of this paper is to provided bounds on the generalization of GNNs in both the in-distribution and out-of-distribution setting. For the in-distribution case, the authors tighten the bounds provided in Liao et al. (2020) by scaling down two separate terms in the PAC-Bayes bound. For the out of distribution setting, the authors analyze the specific issue of size generalization (GNNs trained and tested on subgraphs of differing sizes); the authors provide a bound on the generalization error motivated by the fact that for homophilous graphs, an increase in size does not affect the graph classification with a certain probability.

The setup for the in-distribution generalization error is as follows: there exists a distribution over the set of graphs, node features and graph labels. A GCN is trained on m samples from the distribution, and then the task is to bound the gap between the true risk (probability the graph label is off by a margin gamma) and the empirical risk (the training error).

Liao et al. (2020) provides a bound for the above gap. The present paper tightens two terms. First, the authors claim that the gap does not grow exponentially with the maximum degree but linearly. The authors show that when performing induction over the layers of the GCN, due to normalization, it is possible to maintain a linear dependency on degree (pages 13-14). Second, the authors tighten a separate term by utilizing a random matrix theory theorem from (Vershynin, 2018).

The setup for size generalization (out of distribution) is as follows: there exists a very large graph G. For training, a GCN is trained on subgraphs of G generated by performing random walks of length N and taking the induced subgraph. For testing, however, the test subgraph is generated by a random walk of length M >> N. The goal is still to perform graph classification. Every node in the graph has a label and the subgraph label is simply the most common ground truth or predicted node label.

The core tool needed to bound the size generalization error is the observation that due to homophily it is possible to bound the probability that a random walk of size M reaches a node of a label differing from the label of the initial node. In the case where the random walk does not reach a node of a differing label, the length of the random walk does not impact the subgraph label.

**Summary Of The Review:**

Overall the paper tackles an important GNN issue and moves the community toward understanding the theoretical generalizability of such models. The paper is also comprehensive in considering both in-distribution and out-of-distribution generalization. The largest areas of improvement, in my opinion, are better placing the significance of these results in context and more thorough experiments. The exact benefit of the tightened in-distribution bounds is not clear (would recommend a more detailed experiment isolating the benefit of the linear degree term) and for out-of-distribution, it is not clear when it would be appropriate to sample the large testing subgraph.

---

> ### Author Response · Authors · 2022-11-23
> **Response to Reviewer h6gY**
>
> We thank the reviewer for the positive comments!
>
> > **Q1: It is not fully clear what is the significance / level of contribution of tightening the bounds in Liao et al.”**
>
> A1: Thanks for raising this point! Due to the space limit, we do provide more details and exact bound values in Table 1 and Table 2 in Appendix. From these tables, we can see that our new bound does significantly improve the previous state-of-the-art by orders of magnitudes like 10^3 to 10^6 on synthetic and real-world datasets.
>
> > **Q2: It would be helpful to have more specification for when the expression for M in Theorem 4.3 is compatible with the condition that M >> N.**
>
> A2: Thanks for your suggestion! We added a discussion in the revision. Note that this theorem explicitly constrains M , whereas the only condition on N is that the size-N training error “L_{D_N ,0}” is small. Therefore, our theorem does not exactly require N << M and has quite some flexibility. “N << M” is just a condition that helps motivate our analysis setup. Note that for small N, it is often the case that the size-N training error is small.
>
> > **Q3: Specifically the paper largely side-steps situations in which it is desirable to test the GNN on the large subgraph itself instead of sampling from the test subgraph, for instance the setting considered in “From Local Structures to Size Generalization in Graph Neural Networks”.**
>
> A3: We’d like to clarify that we do test on size-M large subgraphs in our experiments (denoted as “OOD Test acc” in Figure 2 (c) and (f)). We improve the legend of Figure 2 to better visualize it. We also test on randomly sampled size-N subgraphs, which are denoted as “OOD Test acc (sampling)”.
>
> > **Q4: It is helpful that the authors explain the surprising findings in Figures 2c and 2f; however, it would be helpful to also include cases where the test accuracy is not higher than the training accuracy.**
>
> A4: Thanks for raising this good point!
>
> Again, we’d like to clarify that we have two OOD testing accuracies, one is for testing on the size-M large subgraphs and the other is for testing on the randomly sampled size-N subgraphs. The testing performances of size-M subgraphs are indeed worse than training accuracies on synthetic homophilic graphs in Figure 2 (c), whereas the ones for size-N are comparable or slightly better than training. So it is not so surprising on synthetic graphs. As for real-world datasets, since we do not have control over data and label generation, the performance gap could depend on many factors including homophily, labeling generation process, etc.
>
> We also introduced a new experiment involving non-homophilic synthetic graphs in the revision. Like the homophilic synthetic graphs, they are generated with the stochastic block model, but instead with equal probabilities of connection between nodes of the same block and nodes of a different block. Therefore, the node labels among neighboring nodes tend to be uniform, thus being heterophilic. Contrasting with the results shown for the homophilic synthetic graphs, the non-homophilic graph results show that the OOD test accuracies of both sampling size-M and size-N subgraphs are less than the training accuracy in Figure 2(c) (denoted as NonHom), which is expected from our theory. Nonetheless, the performance drop (~5%-10%) is still acceptable since it is just slightly larger than the one in homophilic graphs.

---

### Official Review · Reviewer_Lvuo · 2022-10-25

**Confidence:** 3
**Correctness:** 3
**Technical Novelty And Significance:** 3
**Empirical Novelty And Significance:** 3
**Recommendation:** 6

**Clarity, Quality, Novelty And Reproducibility:**

The clarity and quality of the paper are good, but the novelty seems limited (see comments above). The authors have provided detailed hyper-parameters in the appendix, so the reproducibility should be okay (though providing the source codes will be better).

**Strength And Weaknesses:**

**Pros:**
1. The generalization, specifically out-of-distribution generalization, of GNNs, is an important and trending research direction, and its theoretical analysis is not well studied.
2. The proposed in-distribution bound seems to improve over the existing analysis.

**Cons and questions:**
1. For OOD generalization, this paper focuses on a specific setup, i.e., graph classification where the graphs are sampled from a giant graph, and the graph label is determined by homophily. Though I acknowledge that such an assumption may be unavoidable for rigorous theoretical analysis, it seems to be impractical and thus greatly limits the scope of the proposed analysis. For example, many works on OOD generalization for graph classification focus on molecule classification, where training and testing molecules are collected in different environments/with different backbone structures, which is a completely different scenario and the proposed method seems unable to fit.
2. Following the above comment, the authors observe in experiments that “the GCN model achieves OOD test accuracy on large-subgraph that was comparable to ID accuracy on small-subgraph if not outright better”, which contradicts previous works such as Yehudai et al. (2021). This may also suggest that the assumed setting is not practical.
3. A previous analysis shows that the GNN architectures and downstream graph tasks can greatly affect the generalization of GNNs [1]. I think this work is highly related and a proper discussion should be added.
[1] How Neural Networks Extrapolate From Feedforward to Graph Neural Networks, ICLR 2021.
4. It would also make the paper stronger if the authors can briefly point out how the proposed analysis can inspire improving GNNs, which is crucial for GNN practitioners.

Minor:
(1) Figure 2 is a bit vague (vector graphics are recommended).

===after rebuttal===
I have read the rebuttal and thank the authors for the clarifications. Similar to other reviewers, I think the paper makes interesting theoretical analyeses, but rely on very strong assumptions, so I am not entirely sure whether such contributions meet the bar of ICLR. All things considered, I have increased my score to 6, i.e., slightly positive.

**Summary Of The Paper:**

In this paper, the authors propose theoretical analyses for the generalization of GNNs. For the in-distribution case, the authors propose an improved bound regarding graph classification compared to the existing PAC-Bayes results. For the out-of-distribution generalization, the authors propose an analysis for node classification by using random walks instead of assuming a ground-truth generative model.

**Summary Of The Review:**

This paper proposes theoretical analyses for the generalization of GNNs, which is an important topic. My main concerns lie in the reasonability of the assumptions and the paper's relationships compared to the literature.

---

> ### Author Response · Authors · 2022-11-23
> **Response to Reviewer Lvuo**
>
> We thank the reviewer for the helpful comments!
>
> > **Q1:  Assumptions of OOD generalization.**
>
> A1: We added more discussion and justification in the introduction and Section 4 in the revision. Please see our general response for more details.
>
> We appreciate your understanding that some assumptions may be unavoidable for rigorous theoretical analysis. Since understanding OOD generalization for GNNs is very hard, we need to achieve a tradeoff between assumptions that are practically relevant, and those for which rigorous guarantees are provable.
>
> > **Q2: Contradiction with previous works such as Yehudai et al. (2021) may suggest that the assumed setting is not practical.**
>
> A2: Thanks for raising this point!
>
> We’d like to clarify that we have two OOD testing accuracies, one is for testing on the size-M large subgraphs and the other is for testing on the randomly sampled size-N subgraphs. The testing performances of size-M subgraphs are indeed worse than training accuracies on synthetic homophilic graphs in Figure 2 (c), whereas the ones for size-N are comparable or slightly better than training. So it is not a contradiction to Yehudai et al. (2021) on synthetic graphs. As for real-world datasets, since we do not have control over data and label generation, the performance gap could depend on many factors including homophily, labeling generation process, etc.
>
> On the other hand, the Erdős–Rényi model adopted by Yehudai et al. (2021) is also far from being practical. But both this work and ours shed some light on understanding size generalization for GNNs from different perspectives.
>
> > **Q3: Previous analysis shows that the GNN architectures and downstream graph tasks can greatly affect the generalization of GNNs.**
>
> A3: Thanks for the reference! We cited and included the discussion with this paper in the revision. It focuses on studying the OOD generalization of GNNs in algorithmic tasks (where the target function is assumed) and provides guarantees on extrapolation using NTK. This is still quite different from our generalization error analysis which does not assume anything on target functions and focuses on deriving generalization bounds.
>
> > **Q4: How the proposed analysis can inspire improving GNNs, which is crucial for GNN practitioners.**
>
> A4: Thanks for raising this good point! We believe that (approximately) computing some important graph functions like sparsest cuts would not only provide theoretical guarantees on OOD generalization in certain cases but also serve as good features for GNNs to achieve better OOD performances in the general case. That being said, designing GNNs that could efficiently leverage the spectrum information of the graph Laplacian would be very promising.

---

### Official Review · Reviewer_y9pP · 2022-10-26

**Confidence:** 3
**Correctness:** 3
**Technical Novelty And Significance:** 2
**Empirical Novelty And Significance:** 2
**Recommendation:** 5

**Clarity, Quality, Novelty And Reproducibility:**

Clarify: this paper is well written and organized

Quality: the quality is overall good though I didn't carefully check the proof

Novelty: the algorithmic novelty is limited especially the scope is limited in size generalization which is a particular and simple OOD setting on graphs

**Strength And Weaknesses:**

Pros:

The paper is well motivated and focuses on an important and active research problem in the graph ML community. The theory results seem correct and reasonable though I didn't carefully check the proof in the appendix. The experiment results verify the theoretical argument.

Cons:

1. The analysis is built on several assumptions that may violate the practical settings, like the graph data generation assumption. More discussions and justification are needed.

2. The proposed theory seems to require the homophily assumption of graph structures. How it behaves for heterophilic graphs?

3. The discussed distribution shifts only cover the size variation, which is quite limited in contrast with the various distribution shift types in practice. For example, cross-domain transfer in multi-graph generalization and temporal generalization in dynamic graphs [1], subgroup generalization across majority and minority feature groups [2], motif-structure bias of spurious correlation [3], and substructure-aware distribution shift in molecular property prediction [4], etc. More discussions on how the theory in this paper could shed lights on these practical OOD learning settings can definitely help to strenghthen the paper.

4. The experiments are only conducted on the size generalization task. And, similarly, more experiments to cover the more OOD types, such as the above-mentioned settings, which can be more challenging and closer to the real cases could increase the diversity and strengthen the contributions.

[1] Handling distribution shifts on graphs: an invariance perspective, ICLR22

[2] Subgroup generalization and fairness of graph neural networks, NeurIPS21

[3] Discovering Invariant Rationales for Graph Neural Networks, ICLR22

[4] Learning Substructure Invariance for Out-of-Distribution Molecular Representations, NeurIPS22

**Summary Of The Paper:**

This paper focuses on the generalization ability of graph neural networks and derives the generalization error bound based on PAC-Bayes framework. The new theoretical bound improves the state-of-the-art result, and empirical studies show that the proposed model can help to address size generalization problem on graphs.

**Summary Of The Review:**

Overall, I think this paper is well motivated and written. The theoretical results are interesting and reasonable. However, I believe there still exists much room for improvement based on the current version. For example, more discussions on other out-of-distribution settings and distribution shift types are needed to strengthen the contributions. And, more experiments with other OOD settings could also help to increase the impact of this work.

---

> ### Author Response · Authors · 2022-11-23
> **Response to Reviewer y9pP**
>
> We thank the reviewer for the helpful comments!
>
> > **Q1: Assumptions of OOD generalization.**
>
> A1: We added more discussion and justification in the introduction and Section 4 in the revision. Please see our general response for more details.
>
> > **Q2: How it behaves for heterophilic graphs?**
>
> A2: Thanks for your good suggestion! We included new experiments on heterophilic graphs in the revision (Figure 2 (c) and Appendix B.2.2). Please see our general response for more details.
>
> > **Q3: More discussions on other distribution shifts and how the theory in this paper could shed lights on these practical OOD learning settings can definitely help to strenghthen the paper.**
>
> A3: Thanks for your good suggestion! We actually cited and discussed [2] in the original introduction. For the other references, we cited them and added the discussion in the revision.
>
> > **Q4: More experiments to cover the more OOD types, such as the above-mentioned settings, which can be more challenging and closer to the real cases could increase the diversity and strengthen the contributions.**
>
> A4: We agree that including more OOD types could potentially improve the paper. But it may also defocus our main contribution since each OOD type requires a lengthy explanation of the setup. That is perhaps why each of the suggested references only focuses on one OOD type. We hope to leave it for future work.

---

> > ### Comment · Reviewer_y9pP · 2022-11-28
> > **Thank you for the response and improving the paper**
> >
> > I appreciate the new discussions on related works and extra experiment results, which strengthen this paper a lot.
> >
> > Based on the review comments from other reviewers, I agree that the overclaim regarding the scope and setting is a significant issue. In particular, as revealed by the paper's title which adopts a fairly general scope, the analysis should be built on the general setting and applicable for various OOD types in practice. However, the actual considered setting is a much narrower class and constrains the conclusion to a very specific case, which may limit the significance of this paper at the current point. While I understand that these technical assumptions are often necessary for feasible theoretical analysis especially given that some of them are adopted by some published peer works, I think these premise assumptions and target specific problem setting should be highlighted at the very beginning of this paper, e.g., incorporated into the paper title. This can be really beneficial for better positioning this work with others in this growing and active areas, and also helping the readers to get the key points at the first glance. At the current stage, this submission seems to describe an ambitious big picture and however makes a limited advance in practice.
> >
> > Therefore, considering the above pros and cons, I tend to keep the score as a borderline reject.

---

### Official Review · Reviewer_MGXu · 2022-11-27

**Confidence:** 4
**Correctness:** 2
**Technical Novelty And Significance:** 2
**Empirical Novelty And Significance:** 2
**Recommendation:** 3

**Clarity, Quality, Novelty And Reproducibility:**

This work is well-written and easy to follow. However, many overclaims and misleading experiments, plus the limited scope of the paper, make the novelty of the work limited.

**Strength And Weaknesses:**

While I appreciate the completeness and clarity of the paper, I find the scope of the paper is rather limited, or even overclaimed by the authors.

**Overclaimed improvements in IID generalization bound.**
The authors claimed that reduce an exponential dependency on the node degree to a linear dependency for the IID generalization bound. However, they also exacerbate the dependency on **hidden dimensions $h$** from log scale to linear. This essentially establishes a trade-off between their bound and that from [1]. The advantages only exist for graphs with high degrees and GNNs with deep layers. In contrast, many of realistic graphs (e.g., molecules) tend to have a lower degree. Practitioners tend to adopt a shallower and wider GNN due to the memory cost, especially for homophilic graphs where simple MLP with some post-hoc modifications can achieve top performances (cf. leaderboard results in OGB).

When demonstrating the advances of the established generalization bound over [1], the authors seem to be conducting **misleading comparisons**. Since the advances only exist for graphs with high degrees and GNNs with deep layers, the authors are using a deeper GNN (4, 6, 10 layers compared to 2, 4, 6, 8 layers in [1]), and small hidden dimension (5, 32 compared to 128 in [1]). This makes the improvements and significance of the IID generalization bound limited.

**Overclaims in assumptions for OOD generalization bound.**
When it comes to the OOD generalization bound, the authors claimed their setup has advantages over some OOD setups in the literature where a generative model of graphs and labels is explicitly assumed [2,3,4,5,6,7,8,9]. In particular, in the paragraph of Size Generalization Assumptions in the paper, the authors are essentially making assumptions about the data generation process. The graphs are sampled from random walks with different lengths, which forms a specific graph family just like graphon [2,3,6,8,9]. The labels are determined by the majority of labels in the training graphs, which essentially have little difference from the causal assumptions made in [2,3,4,5,6,7,8]. I didn’t see the advantages of the assumptions made in the paper.

However, this paper introduces additional assumptions that require the graphs to be homophilic, which makes the random walk sampling over the large graphs trivial. As [10] already found that random walks with more steps will converge to some stationary distributions over the original graphs. Therefore, analyzing homophilic graphs sampled using random walks with more steps seems to be less interesting.


**Limited scope and poor coverage of the literature.**
Although, as pointed out by other reviewers that this work is limited to graph size shifts and missed discussion with many related works, I still find many missing discussions and comparisons in [2,3,4,5,6,7,8,9]. In particular, [2,3,6,8] studied graph size shifts as well, but I can’t find any discussions in work. Both the theoretical and empirical parts of the work lack a comparison with these works.

References:

[1] A PAC-Bayesian approach to generalization bounds for graph neural networks, ICLR21.

[2] From Local Structures to Size Generalization in Graph Neural Networks, ICML21.

[3] Size-Invariant Graph Representations for Graph Classification Extrapolations, ICML21.

[4] Handling distribution shifts on graphs: an invariance perspective, ICLR22.

[5] Discovering Invariant Rationales for Graph Neural Networks, ICLR22.

[6] Invariance Principle Meets Out-of-Distribution Generalization on Graphs, ICML22: Workshop on Spurious Correlations, Invariance and Stability.

[7] Learning Substructure Invariance for Out-of-Distribution Molecular Representations, NeurIPS22.

[8] OOD Link Prediction Generalization Capabilities of Message-Passing GNNs in Larger Test Graphs, NeurIPS22.

[9] Generalization Analysis of Message Passing Neural Networks on Large Random Graphs, arXiv22.

[10] Representation Learning on Graphs with Jumping Knowledge Networks, ICML18.


**Summary Of The Paper:**

This paper studies the PAC-Bayes generalization bound for both IID and OOD generalization on graphs, with a focus on homophilic graphs and graph size shifts. In particular, the authors reduce an exponential dependency on the node degree to a linear dependency for the IID generalization bound. Then they further apply the generalization bound to study random walk sampled graphs with different sizes. Besides, they conduct some experiments to support the derived bounds.

**Summary Of The Review:**

Although I appreciate the completeness and clarity of the paper, many overclaims and misleading experiments, plus the limited scope of the paper, make the paper a clear reject.

---

> ### Author Response · Authors · 2022-12-01
> **Response to Reviewer MGXu**
>
> Thanks for the comments!
>
> > **Q1: Overclaimed improvements in IID generalization bound.**
>
> A1: We believe the reviewer has some misunderstanding about our contributions on the in-distribution (ID) generalization bound. In particular, the essential improvement between new vs. old bounds is $d(h + ln(l))$ vs. $d^{l-1} h ln(hl)$, where $d$ is the max node degree, $l$ is the depth, and $h$ is the hidden dimension. When the depth $l$ increases, the growth of the bound value is mainly governed by the exponential term as indicated in our experiments. Therefore, the empirical improvement is indeed significant.
>
> As for the extreme shallow case where $l=2$, i.e., only one hidden layer, these two bound values become almost indistinguishable in the log domain since $d^{l-1} = d$. However, this case is not that interesting since people seldom use one-hidden-layer GCN in practice. In contrast, people care more about how deep graph networks generalize.
>
> At last, we performed experiments when $l=3$ (seen to be used in practice) and include the results (**natural logarithm values**) below.
> | Dataset     | ER-1 | ER-2 | ER-3 | ER-4 | SBM-1 | SBM-2 | IMDBBINARY | IMDBMULTI | PROTEINS | COLLAB |
> | -------- | ---- | ---- | ---- | ---- | ------- | ------- | ---------- | --------- | -------- | ------ |
> | Old Bound   |12.23 |12.87 |13.21 |13.43 | 12.80 | 12.56 | 13.33 | 12.93 | 10.83 | 15.28 |
> | New bound   |**10.60**|**10.92**|**11.08**|**11.19**|**10.88**|**10.75**|**10.88**|**10.69**|**9.20**|**12.18**|
>
> It is clear that our new bound does bring orders of magnitudes of improvement on all synthetic and real-world datasets.
>
> > **Q2: In particular, in the paragraph of Size Generalization Assumptions in the paper, the authors are essentially making assumptions about the data generation process. The graphs are sampled from random walks with different lengths, which forms a specific graph family just like graphon [2,3,6,8,9].**
>
> A2: We'd like to clarify that we do not assume how the single large graph is generated. But we assume subgraphs are generated from the large graph following random walks, which are valid assumptions since many practical works do use that to collect subgraphs [1,2,3,4,5].
>
> Moreover, we disagree with your argument that our setup forms a specific graph family just like graphon. **Our setup forms a specific graph family, which is not like graphon and is more flexible**. For graphons, all pairs of nodes share the same kernel to determine the edge probability. For OOD analysis with graphons, this kernel is assumed to be the same between training and testing graphs. These properties/assumptions are quite restrictive in practice. We neither assume anything on the edge probability in the underlying large graph nor the shared kernel between training and testing subgraphs.
>
> [1] Hamilton, W., Ying, Z. and Leskovec, J., 2017. Inductive representation learning on large graphs. Advances in neural information processing systems, 30.
>
> [2] Grover, A. and Leskovec, J., 2016, August. node2vec: Scalable feature learning for networks. In Proceedings of the 22nd ACM SIGKDD international conference on Knowledge discovery and data mining (pp. 855-864).
>
> [3] Perozzi, B., Al-Rfou, R. and Skiena, S., 2014, August. Deepwalk: Online learning of social representations. In Proceedings of the 20th ACM SIGKDD international conference on Knowledge discovery and data mining (pp. 701-710).
>
> [4] Gärtner, T., Flach, P. and Wrobel, S., 2003. On graph kernels: Hardness results and efficient alternatives. In Learning theory and kernel machines (pp. 129-143). Springer, Berlin, Heidelberg.
>
> [5] Borgwardt, K.M., Ong, C.S., Schönauer, S., Vishwanathan, S.V.N., Smola, A.J. and Kriegel, H.P., 2005. Protein function prediction via graph kernels. Bioinformatics, 21(suppl_1), pp.i47-i56.
>
> >**Q3: Limited scope and poor coverage of the literature. In particular, [2,3,6,8] studied graph size shifts as well, but I can’t find any discussions in work.**
>
> A3: **We already discussed [2] in the original version. Please see the beginning of the OOD Generalization paragraph in the introduction.** For the 10 references you mentioned, we actually discussed 6 of them already in the paper. As for the other references you suggested like [3, 6, 8], none of them analyze the OOD generalization bound for GNNs, thus being indirectly relevant. Moreover, they all belong to a class of methods that assume causal models and aim for certain invariances. We already discussed this class of methods in the revised version. We could include more discussion in the future. But the argument about the poor coverage of the literature is unwarranted.

---

> > ### Comment · Reviewer_MGXu · 2022-12-02
> > **Response to Authors’ rebuttal**
> >
> > Thanks for the response. My biggest concern is the overclaims made in this work. From the author's response, I couldn’t find any ***factual evidence*** to justify the overclaims.
> >
> > **Comment on A1:**
> > From the authors’ reply, I am more confident about the overclaim on the significance of the generalization bound. As I mentioned in the review, it is essentially a trade-off. On the one hand, it decreases the exponential dependency on the node degree. However, it exacerbates the dependency on **hidden dimensions h**. All of the current experiments are biased and focus on the number of layers **l** but neglect the exacerbated dependency on the hidden dimension **h**. If **h** is also considered in comparison with [1], the improvements should not be as significant as claimed by the authors.
> >
> > **Comment on A2:**
> > I can’t see any flexibility nor any significance to previous works as listed in my review.
> >
> > In particular, the subgraphs are sampled using random walks with different lengths on homophilic graphs, and the labels are determined by the majority of the node labels. These assumptions also impose additional constraints on the data generation process. The additional assumptions made in this paper, as pointed out by other reviewers, can be too strong for practical use.
> >
> > Besides, a similar analysis has already been conducted in the literature. For example, Xu et al. already found that random walks with more steps will converge to some stationary distributions over the original graphs. Therefore, analyzing homophilic graphs sampled using random walks with more steps seems to be less interesting.
> >
> > **Comment on A3:**
> > To be more clear: the limited coverage refers to both related work discussions and experiments.
> >
> > First, the mentioned references all provide theoretical justifications for the OOD generalization ability under multiple distribution shifts on graphs, including the size shifts analyzed in the paper. In other words, they are all closely relevant works.
> >
> > If the proposed approaches in these works already mitigated the distribution shifts analyzed in the paper theoretically or empirically (or not), the derived results of this work should draw some correspondence to better declare the position of these results in the literature.

---

> > > ### Author Response · Authors · 2022-12-08
> > > **Can you explain how our results exacerbate the dependency on h?**
> > >
> > > Please read our paper carefully before criticizing our in-distribution contribution as overclaimed!
> > >
> > > We have made an improvement on the dependency of max node degree $d$ and hidden dimension $h$ by changing the dependency from $\sqrt{d^{l-1} h \ln(lh)}$ to $\sqrt{d (h + \ln(l))}$.
> > > **This is a strict improvement as long as l>1 and h>1**!
> > >
> > > Can you explain how our results exacerbate the dependency on h?

---

> > > ### Author Response · Authors · 2022-12-09
> > > **Response to Comment on A2**
> > >
> > > > Besides, a similar analysis has already been conducted in the literature. For example, Xu et al. already found that random walks with more steps will converge to some stationary distributions over the original graphs.
> > >
> > > This is a response to a claim that we are not making. Our analysis is not making claims about the behavior of long-term random walks and convergence, rather, we provide probability bounds to understand the predictive power of short random walks.

---

### Author Response · Authors · 2022-11-23
**General Response From Authors**

We thank all reviewers for the valuable feedback and suggestions. We address most of the questions and suggestions in the revised manuscript (**marked in blue color**).

First, we’d like to restate our two key contributions:

1. We improve upon the state-of-the-art PAC-Bayes (in-distribution) generalization bound (reducing an exponential dependency on the node degree to a linear dependency)
2. Relying on spectral graph theory, we prove some rigorous guarantees about the out-of-distribution (OOD) size generalization of GNNs.

We now respond to some common questions.

> **Q1: Assumptions of OOD generalization.**

A1: **We have revised the “OOD Generalization” portion of page 2 to discuss related work, and expanded the first paragraph of Section 4 to explain the reasoning.**

Since we adopt a statistical learning viewpoint, there must necessarily be some assumptions relating the training and testing graphs (otherwise the No-Free Lunch theorem applies). There is a tradeoff between assumptions that are practically relevant, and those for which rigorous guarantees are provable.

In particular, we focus on the situation where an extremely large graph is presented and the training and testing graphs are sampled from it. This is indeed a typical scenario happening on large social networks and has been studied in, e.g., [1,2]. Moreover, random walk based graph representation learning has a long history [3,4,5], which supports our motivation.

Moreover, in contrast to previous theoretical works which mostly assume a known family of graph generative models like the Erdős–Rényi model or Graphons, our analysis avoids such an assumption since it violates the tradition in statistical learning theory where the underlying data distribution is unknown.

We agree that our analysis setup would not cover certain applications like molecule classification. However, we do not think this undermines our contributions. The reasons are as follows. 1) OOD generalization is hard in practice and people are exploring different techniques in different domains/problems, thus being somewhat case-by-case. 2) The theoretical OOD generalization analysis is even harder since no known analysis framework works for most problems. We believe our work makes progress on a fairly large class of problems.

[1] Hamilton, W., Ying, Z. and Leskovec, J., 2017. Inductive representation learning on large graphs. Advances in neural information processing systems, 30.

[2] Grover, A. and Leskovec, J., 2016, August. node2vec: Scalable feature learning for networks. In Proceedings of the 22nd ACM SIGKDD international conference on Knowledge discovery and data mining (pp. 855-864).

[3] Perozzi, B., Al-Rfou, R. and Skiena, S., 2014, August. Deepwalk: Online learning of social representations. In Proceedings of the 20th ACM SIGKDD international conference on Knowledge discovery and data mining (pp. 701-710).

[4] Gärtner, T., Flach, P. and Wrobel, S., 2003. On graph kernels: Hardness results and efficient alternatives. In Learning theory and kernel machines (pp. 129-143). Springer, Berlin, Heidelberg.

[5] Borgwardt, K.M., Ong, C.S., Schönauer, S., Vishwanathan, S.V.N., Smola, A.J. and Kriegel, H.P., 2005. Protein function prediction via graph kernels. Bioinformatics, 21(suppl_1), pp.i47-i56.

> **Q2: How does theory behave on heterophilic graphs?**

A2: **We have introduced a new experiment involving non-homophilic synthetic graphs**. Like the homophilic synthetic graphs, they are generated with the stochastic block model, but instead with equal probabilities of connection between nodes of the same block and nodes of a different block. Therefore, the node labels among neighboring nodes tend to be uniform, thus being heterophilic. Contrasting with the results shown for the homophilic synthetic graphs, the non-homophilic graph results show that the OOD test accuracies of both sampling size-M and size-N subgraphs are less than the training accuracy in Figure 2(c) (denoted as NonHom), which is expected from our theory. Nonetheless, the performance drop (~5%-10%) is still acceptable since it is just slightly larger than the one in homophilic graphs.

---

### Decision · Program_Chairs · 2023-01-20

**Decision:**

Reject

**Justification For Why Not Higher Score:**

According to my expertise and reviewing process, this paper should belong to a Reject.

**Justification For Why Not Lower Score:**

According to my expertise and reviewing process, this paper should belong to a Reject.

**Metareview: Summary, Strengths And Weaknesses:**

The paper describes in-distribution and out-of-distribution generalization with graph convolutional networks. In the former setting, the paper improves on an existing bound in the literature by reducing dependency on maximum node degree to a linear factor. In the latter setting, the paper builds on in-distribution results, and considers the case of homophilic and random walk sampled graph structures. Meanwhile, the paper conducts an empirical analysis on a variety of synthetic and real-world datasets, showing performance levels consistent with the theory, and even demonstrating strong performance when certain assumptions are violated (degree regularity for example). Therefore, the paper makes a useful contribution for studying the generalization performance of GNNs.

This paper has some merits, and I personally like the idea of this paper. However, most of reviewers and I have some concerns about the assumptions for OOD generalization bound. When deriving the OOD generalization bound, the authors claimed their setup has advantages over some OOD setups in the literature where a generative model of graphs and labels is explicitly assumed. Yet the assumptions could be too strong to guidance the practice. The considered distribution shifts are limited to graph size shifts as well. According to the top-tier conference criteria in the ML community, this is a boarderline paper with mixed opinions. The next version must be a strong paper if authors can take comments into consideration.

**Summary Of Ac-Reviewer Meeting:**

After discussing with Reviewers via zoom, most of reviewers and I have some concerns about the assumptions for OOD generalization bound. When deriving the OOD generalization bound, the authors claimed their setup has advantages over some OOD setups in the literature where a generative model of graphs and labels is explicitly assumed. Yet the assumptions could be too strong to guidance the practice. The considered distribution shifts are limited to graph size shifts as well. According to the top-tier conference criteria in the ML community, this is a boarderline paper with mixed opinions. The next version must be a strong paper if authors can take comments into consideration.